# Transcranial direct current stimulation of cerebellum alters spiking precision in cerebellar cortex: A modeling study of cellular responses

Xu Zhang[1,2,3], Roeland Hancock[2,3,4], Sabato Santaniello[1,2,3]*

**1** Biomedical Engineering Department, University of Connecticut, Storrs, Connecticut, United States of America, **2** Brain Imaging Research Center, University of Connecticut, Storrs, Connecticut, United States of America, **3** Connecticut Institute for the Brain and Cognitive Sciences, University of Connecticut, Storrs, Connecticut, United States of America, **4** Department of Psychological Sciences, University of Connecticut, Storrs, Connecticut, United States of America

* sabato.santaniello@uconn.edu

**Data Availability Statement:** Code and numerical simulation scripts are available on ModelDB at the URL: http://modeldb.yale.edu/267189.

## Abstract

Transcranial direct current stimulation (tDCS) of the cerebellum has rapidly raised interest but the effects of tDCS on cerebellar neurons remain unclear. Assessing the cellular response to tDCS is challenging because of the uneven, highly stratified cytoarchitecture of the cerebellum, within which cellular morphologies, physiological properties, and function vary largely across several types of neurons. In this study, we combine MRI-based segmentation of the cerebellum and a finite element model of the tDCS-induced electric field (EF) inside the cerebellum to determine the field imposed on the cerebellar neurons throughout the region. We then pair the EF with multicompartment models of the Purkinje cell (PC), deep cerebellar neuron (DCN), and granule cell (GrC) and quantify the acute response of these neurons under various orientations, physiological conditions, and sequences of presynaptic stimuli. We show that cerebellar tDCS significantly modulates the postsynaptic spiking precision of the PC, which is expressed as a change in the spike count and timing in response to presynaptic stimuli. tDCS has modest effects, instead, on the PC tonic firing at rest and on the postsynaptic activity of DCN and GrC. In Purkinje cells, anodal tDCS shortens the repolarization phase following complex spikes (-14.7 ± 6.5% of baseline value, mean ± S.D.; max: -22.7%) and promotes burstiness with longer bursts compared to resting conditions. Cathodal tDCS, instead, promotes irregular spiking by enhancing somatic excitability and significantly prolongs the repolarization after complex spikes compared to baseline (+37.0 ± 28.9%, mean ± S.D.; max: +84.3%). tDCS-induced changes to the repolarization phase and firing pattern exceed 10% of the baseline values in Purkinje cells covering up to 20% of the cerebellar cortex, with the effects being distributed along the EF direction and concentrated in the area under the electrode over the cerebellum. Altogether, the acute effects of tDCS on cerebellum mainly focus on Purkinje cells and modulate the precision of the response to synaptic stimuli, thus having the largest impact when the cerebellar cortex is active. Since the spatiotemporal precision of the PC spiking is critical to

**Funding:** This work was partly supported by the US NSF (National Science Foundation) CAREER Award 1845348 to S.S. The funders had no role in study design, data collection and analysis, decision to publish, or preparation of the manuscript.

learning and coordination, our results suggest cerebellar tDCS as a viable therapeutic option for disorders involving cerebellar hyperactivity such as ataxia.

## Author summary

Transcranial direct current stimulation (tDCS) of the cerebellum is gaining momentum as a neuromodulation tool for the treatment of neurological diseases like movement disorders. Nonetheless, the response of cells in the cerebellum to tDCS is unclear and hardly generalizes from our understanding of tDCS of the cerebral cortex. We use computational models to investigate the response of several types of cerebellar neurons to the electric field induced by tDCS and show that, differently from the cerebral cortex, tDCS has significant acute effects on the cerebellar cortex. These effects (i) primarily alter the way Purkinje cells encode synaptic stimuli from the molecular layer and (ii) can help hyperactive cells regain postsynaptic spiking precision. Since the spatiotemporal precision of the Purkinje cell spiking is critical to learning and coordination, the study shows how tDCS can operate at the cellular level to treat movement disorders like tremor and ataxia.

## 1. Introduction

The cerebellum has become a target for transcranial direct current stimulation (tDCS) and holds promise for both basic research [1] and treatment of movement disorders [2,3]. The premise underlying clinical tDCS applications is derived from early studies involving the cerebral motor cortex [4,5]. These studies showed that tDCS can evoke sustained polarization of the neurons in the targeted area, which can result in polarity- and time-dependent neuroplastic effects that outlast the stimulation period. However, the distinct cytoarchitecture of cerebellum, which involves numerous folded lobules and gyri arranged in a highly stratified fashion [6], prevents direct translation of cerebral tDCS results [7]. Also, studies of cerebellar tDCS have been characterized thus far by significant methodological variations regarding montages and physiological tasks, which have often resulted in divergent conclusions about the effects of tDCS on the neuronal excitability in the cerebellum, e.g., [8] versus [9]. Accordingly, there has been limited consensus about the nature, magnitude, and relevance of the physiological effects evoked by cerebellar tDCS [7,10].

*In vitro* studies of tDCS using hippocampal slices have shown that extracellular uniform DC electric fields (EF) can differentially polarize axonal or dendritic compartments, depending on the orientation of these compartments with respect to the field [11–13], and enhance synaptic gain and cooperativity [14]. However, in these studies the cellular responses were observed under much higher EF intensities (i.e., $\geq$20 V/m) than the intracranial EF generated *in vivo* by tDCS (typically <2 V/m, [15,16]). Hence, it is unknown whether the effects reported for high-intensity EF also persist under low-intensity tDCS-evoked EF. Also, as tDCS is expected to shift the neuronal membrane potential by less than 1 mV [11], it is unclear whether and to what extent such weak polarization can affect specific cellular compartments or modulate the overall behavior of neurons [17].

To address these questions, we used computational modeling to (i) estimate the effects of tDCS-induced EF on cerebellar neurons and (ii) determine how compartment-specific polarization alters the spiking pattern of these neurons. The cerebellum is an ideal target to study EF-induced modulation of spiking patterns because the cerebellar cortex has a highly

stereotypical spatial organization, and the physiological properties of the cerebellar neuron types (e.g., Purkinje cells, granule cells, Golgi cells, and interneurons) have mostly been identified [18–20]. Also, recurrent connections are less frequent in cerebellar circuits than cerebral circuits [21], which may ease isolating the contribution of individual neuron types to the overall effects of cerebellar tDCS. Finally, Purkinje cells, which are by far the largest neuron type in the cerebellum and therefore a likely contributor to the cerebellar response to tDCS [13,22], exhibit significantly different spiking patterns *in vivo* compared to *in vitro* (e.g., simple spiking, complex spiking, or spontaneous bursting [23–25]), and such patterns depend on the underlying physiological state. Hence, studying the cellular response to cerebellar tDCS can help determine whether tDCS exerts differential effects on neurons depending on the underlying physiological state.

In this study, we combined MRI scans of the human cerebellum and finite-element modeling of the EF induced by tDCS at 2 mA. We paired the EF with 3D multicompartment models of neurons from the cerebellar cortex (Purkinje and granule cells) and the dentate nucleus (deep cerebellar neurons), and we investigated the acute response to synaptic stimuli elicited under different EF orientations. We report that the Purkinje cells are the most sensitive cell type to tDCS, with significant changes in excitability during simple spiking. Large variations were also observed for the Purkinje cell's burst rate, burst duration, and the repolarization phase following complex spiking. Our results suggest that even weak EF as those generated by cerebellar tDCS in realistic *in vivo* settings can conspicuously modulate the cellular behavior, which may lead to functional changes of physiological activity.

## 2. Results

### 2.1 Cerebellar tDCS induces mild electric fields across cerebellum

We considered four 3T MRI brain scans of human subjects from [26] (atlas 1, 2, 4, and 5; voxel dimension: 600 µm) and, for each subject, we used the ROAST software suite [27] to segment the images and estimate the spatial distribution of the EF, $\overrightarrow{E} = \nabla v$, induced by 2-mA-cerebellar tDCS (see definition of $v$ in Methods section). We considered two electrode configurations, i.e., regular (R)-tDCS with two pad electrodes (anode and cathode; size: 50 mm×30 mm×3 mm, **Fig 1A (i)**) and high density (HD)-tDCS with five ring electrodes (1 anode and 4 cathodes; inner radius: 4 mm; outer radius: 6 mm; thickness: 2 mm, **Fig 1B (i)**). We compared R-tDCS versus HD-tDCS in terms of direction, magnitude, and spatial distribution of the polarization induced in the cerebellum.

During the segmentation process, different conductivities were used for six tissue types (i.e., gray matter, white matter, CSF, skull, skin, and air, see **S1 Table**), with values 0.126 S/m and 0.276 S/m assigned to white matter, including the cerebellar cortices, and gray matter, including the cerebellar nuclei, respectively. After estimating the EF intensity and orientation across the entire brain, we then focused on the cerebellar regions, which were manually segmented by Park *et al.* [26]. Segmentation masks were obtained from http://cobralab.ca/atlases/Cerebellum/.

**Fig 1A** and **1B (ii)-(iv)** report the spatial arrangement of the EF along a coronal (ii), sagittal (iii), and axial (iv) section in one subject (atlas 5) for R-tDCS and HD-tDCS, respectively. The sample distribution of the EF intensity estimated for R-tDCS and HD-tDCS in the cerebellar regions of atlas 5 is reported in **Fig 2A and 2B**, respectively. We estimated the EF intensity distribution separately for voxels spanning the left hemisphere, on which the tDCS anode was placed, and the right hemisphere, and we distinguished between voxels spanning the cerebellar cortices and nuclei. **S1A–S1C Fig** report the EF intensity distribution for R-tDCS applied on the remaining subjects (i.e., atlas 1, 2, and 4, respectively).

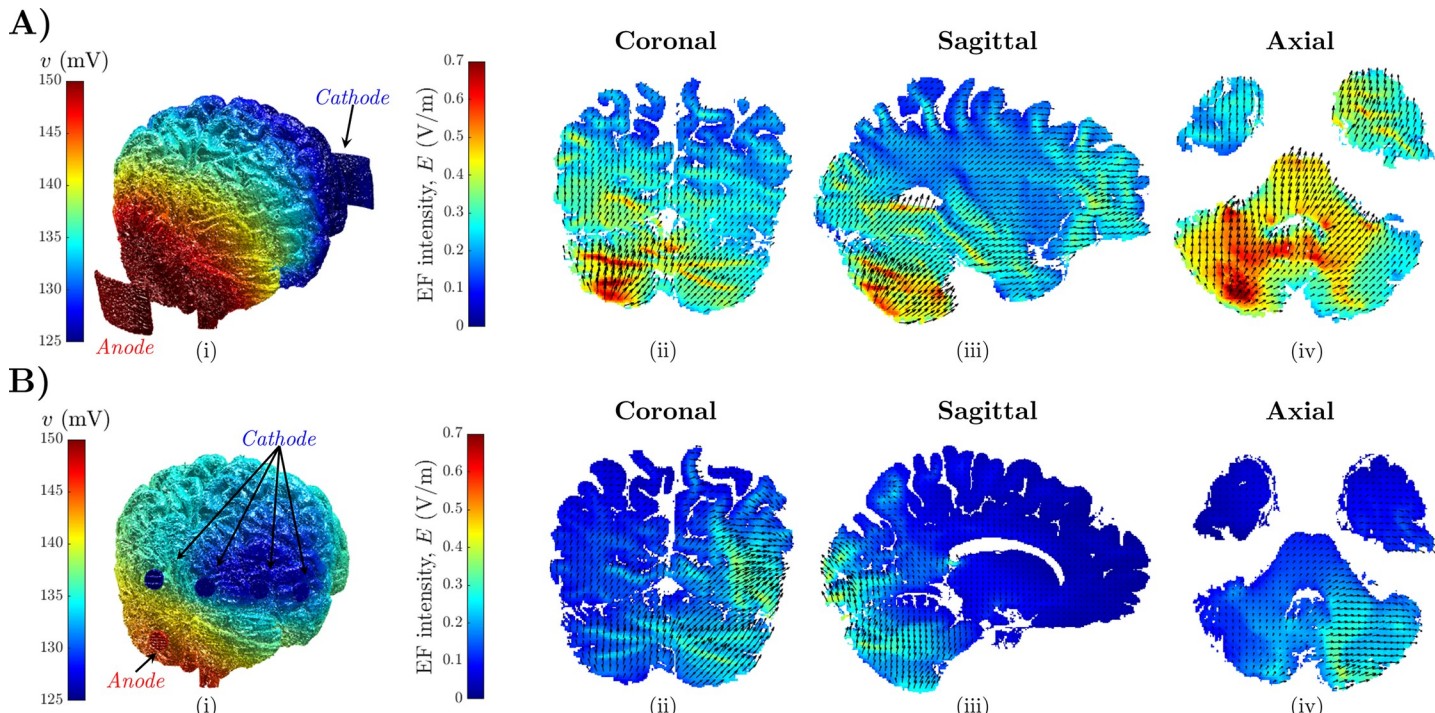

**Fig 1. Estimation of the intensity and orientation of the electric field (EF).** (A-B) Estimated intensity and orientation of the EF induced by 2-mA-cerebellar regular tDCS (A) and high-density tDCS (B) in the brain for one human subject, i.e., atlas 5 from [26]. Panels (i) report the position of the anode and cathode for regular tDCS (two pad electrodes) and high-density tDCS (5 ring electrodes), respectively. Panels (ii), (iii), and (iv) report a coronal, sagittal, and axial view of the EF distribution in atlas 5 for regular (A) and high-density (B) 2-mA tDCS, respectively. Colormap in (i) indicates the distribution of the electric potential, $v$ (scale on the left). In (ii)-(iv), color scales on the left indicate the EF intensity, and black arrows indicate the EF orientation.

In case of R-tDCS, we found that, across all subjects, a significant portion of the cerebellum under the anode (i.e., source electrode on the left hemisphere) received EF at intensity between 0.4 V/m and 0.5 V/m, which is consistent with previous studies conducted using SimNIBS software [28,29]. The EF intensity was, on average, higher in the cerebellar nuclei (0.46 ± 0.07 V/m, mean ± S.D. across four atlases) compared to the cerebellar cortex (0.45 ± 0.34 V/m; one-way ANOVA test after Bonferroni correction, $P$-value $P<0.001$). However, high EF intensity values (i.e., 1 V/m or above) were found in cerebellar cortex (1.29% of the assigned volume) under the targeted (left) hemisphere but not in the cerebellar nuclei.

The electric field also spilled over the untargeted right hemisphere, **Fig 1A and 1B (ii)-(iv)**. For R-tDCS, EF intensity in the right cerebellar nuclei was comparable to the intensity in the left nuclei (0.43 ± 0.06 V/m, mean ± S.D. across four atlases), **Fig 2A (i)-(ii)**; vice versa, EF intensity in the right cerebellar cortex was significantly lower compared to the contralateral cortex (0.34 ± 0.17 V/m, mean ± S.D.; one-way ANOVA test after Bonferroni correction, $P<0.001$). Also, on the left hemisphere, EF intensity exceeded 1 V/m and 1.5 V/m in 1.27% and 0.8% of the cerebellar cortex, respectively. On the right hemisphere, instead, EF intensity exceeded 1 V/m and 1.5 V/m in just 0.3% and 0.12% of the cortex, respectively. No voxel spanning cerebellar nuclei received EF intensity at 1 V/m or above.

Compared to R-tDCS, the five-electrode HD-tDCS montage had the anode in a medial position between the two cerebellar hemispheres, **Fig 1B (i)**, and this resulted in a significant reduction of the EF intensity across both hemispheres, **Fig 2B (i)-(ii)**. HD-tDCS-induced EF had lower intensity compared to R-tDCS both in the cerebellar cortices and nuclei and presented no significant trend when transitioning from the target left hemisphere to the right

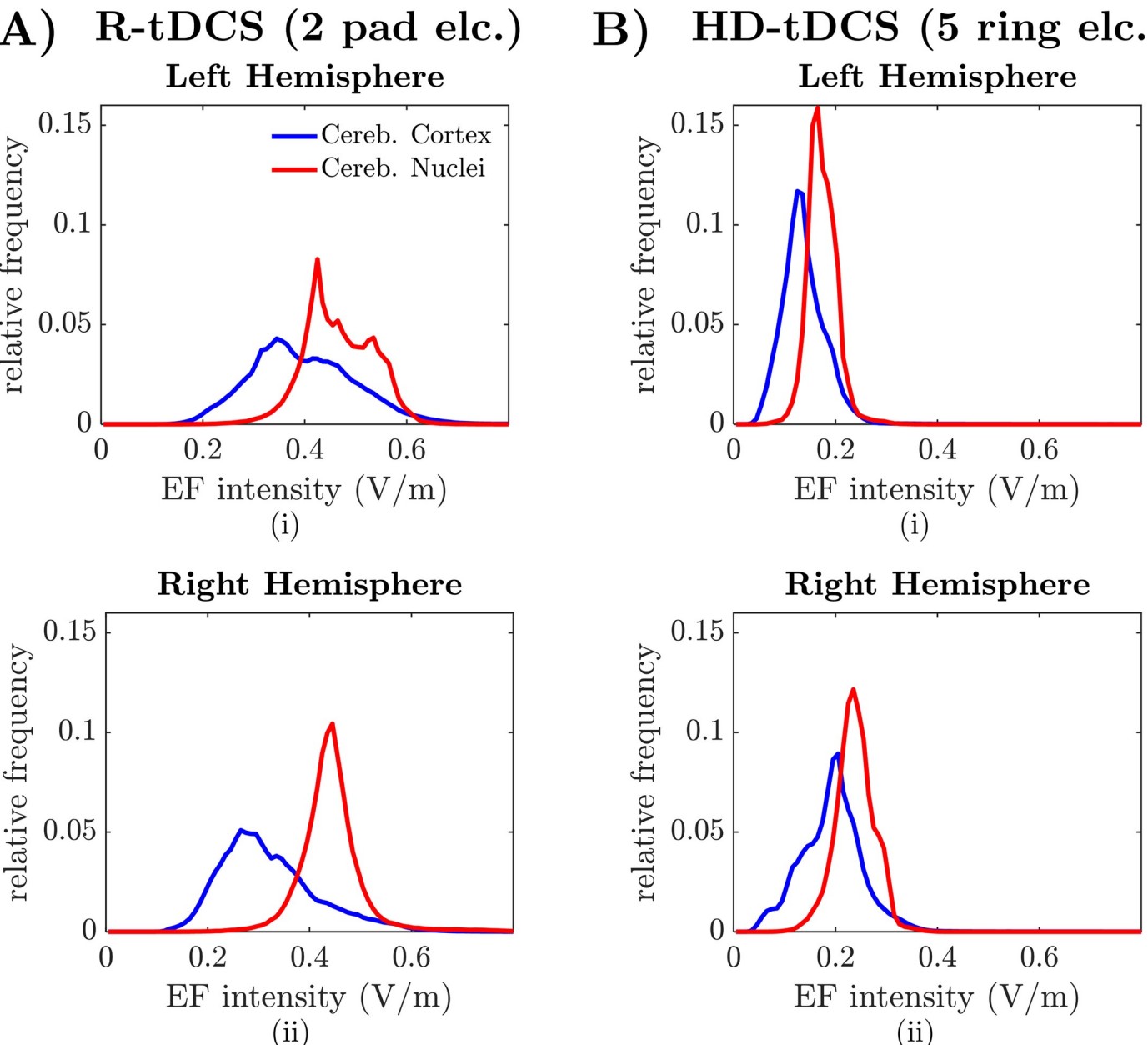

**Fig 2. Distribution of the EF intensity across the cerebellar volume.** (A-B) Sample EF intensity probability distribution estimated for the cerebellar hemispheres in one human subject, i.e., atlas 5 from [26], under a 2-mA cerebellar R-tDCS with two pad electrodes (A) and HD-tDCS with five ring electrodes (B), respectively. Panels (i)-(ii) report the EF intensity distribution estimated for voxels mapping the left hemisphere and right hemisphere, respectively. In both hemispheres, the EF intensity probability distribution is estimated separately for voxels mapping the cerebellar cortex (*blue lines*) and the cerebellar nuclei (*red lines*). Anode was placed on the left hemisphere for both R-tDCS and HD-tDCS.

hemisphere (left nuclei: 0.17 ± 0.029 V/m; left cortex: 0.14 ± 0.067 V/m; right nuclei: 0.24 ± 0.038 V/m; right cortex: 0.20 ± 0.075 V/m, mean ± S.D. across four atlases). The average EF intensity across all cerebellar tissues was approximately 49% lower under HD-tDCS compared to R-tDCS (0.20 ± 0.15 V/m versus 0.40 ± 0.27 V/m, respectively), and this remained consistent both for cerebellar nuclei and cortices (one-way ANOVA test after Bonferroni correction, *P*<0.001).

To further determine whether the differences between R-tDCS and HD-tDCS were influenced by the number and position of the electrodes or the size and shape of the electrodes, we considered the R-tDCS configuration and replaced the pad electrodes with two ring electrodes of the same size as the electrodes used for HD-tDCS, **S2A (i) Fig**. We found that the modified R-tDCS with ring electrodes produce an EF that is similar in shape to the EF elicited by R-tDCS with pad electrodes, **S2A (ii)-(iv) Fig**. The EF generated with ring electrodes, though, was both more spatially limited and of weaker intensity across the cerebellum compared to R-tDCS with pad electrodes. The EF intensity, in fact, was reported at $0.35 \pm 0.05$ V/m (cerebellar nuclei) and $0.31 \pm 0.19$ V/m (cortex) in the left hemisphere and $0.34 \pm 0.05$ V/m (cerebellar nuclei) and $0.24 \pm 0.09$ V/m (cortex) in the right hemisphere (mean $\pm$ S.D.), respectively. Also, in case of R-tDCS with ring electrodes, EF intensity across the cerebellar cortex was significant different in the left hemisphere versus the right hemisphere (one-way ANOVA test after Bonferroni correction, $P < 0.001$), **S2B (i)-(ii) Fig**.

Altogether, these results indicate that cerebellar R-tDCS with pad electrodes produced the strongest and most diffuse EF within the cerebellum, with modest spillover from the target hemisphere to the contralateral hemisphere. EF intensity, though, remained below 1.5 V/m in over 99% of the cerebellar volume, which is consistent with earlier studies [15,16,30] and suggests that tDCS imposes an EF on cerebellar neurons that is mild and below regulatory safety limits [31].

## 2.2 Cerebellar tDCS induces mild polarization of cerebellar neurons

We investigated the effect of applied EF of various intensity and orientation on multicompartment models of the Purkinje cell (PC), granule cell (GrC), and deep cerebellar neuron (DCN), **Fig 3A (i)-(iii)**. The maximum EF intensity (i.e., 1.5 V/m) was obtained from the estimated EF induced in cerebellum under R-tDCS with pad electrodes, **Fig 1A**. For each neuron type, the EF orientation was defined with respect to the neuron's somato-axonal axis and expressed by angle $\theta$, **Fig 3A (i)**. As in [11–13,32], EF pointing towards the axonal terminal of the neuron were defined "*anodal*" and assigned negative intensity. EF pointing towards the dendritic tree, instead, were defined "*cathodal*" and assigned positive intensity. EF intensity of 0 V/m indicates no stimulation.

Neuron models under EF-induced polarization were simulated in NEURON [33]. The PC and DCN models were presented in [34] and [35], respectively. The GrC model was developed for this study by adding the Hodgkin-Huxley models of ionic channels proposed in [36] to the morphology of a granule cell from an adult human subject (cell NMO_32569) from NeuroMorpho.Org [37], **Fig 3A (ii)**. Electrotonic properties and maximum ionic conductance values of the GrC model are reported in **S2** and **S3 Tables**, respectively. Other cerebellar neuron types (e.g., Golgi cells and interneurons) were omitted in this study because they are significantly less numerous and more sparsely distributed compared to PC, DCN, and GrC.

For each neuron type, we adopted the quasi-uniform assumption [38] and calculated the EF-induced polarization as proposed in [39–41]. Specifically, for any neuron compartment, $n$, the extracellular potential, $V_{e,n}$, was assigned as $V_{e,n} = V_{e,0} + \ell_n E$, where $V_{e,0}$ is the extracellular potential of a reference node (i.e., the soma compartment), $E$ is the EF intensity, and $\ell_n$ is the distance of the compartment to the reference node along the EF direction. Each neuron compartment was assigned an extracellular potential via the **extracellular()** mechanism in NEURON.

As in previous studies [42], a linear polarization profile was observed along the soma and dendrites of PC model, **Fig 3B (i)**, and a similar trend was observed for the GrC and DCN models. When applied to the neuron membrane, though, the EF-induced polarization profile

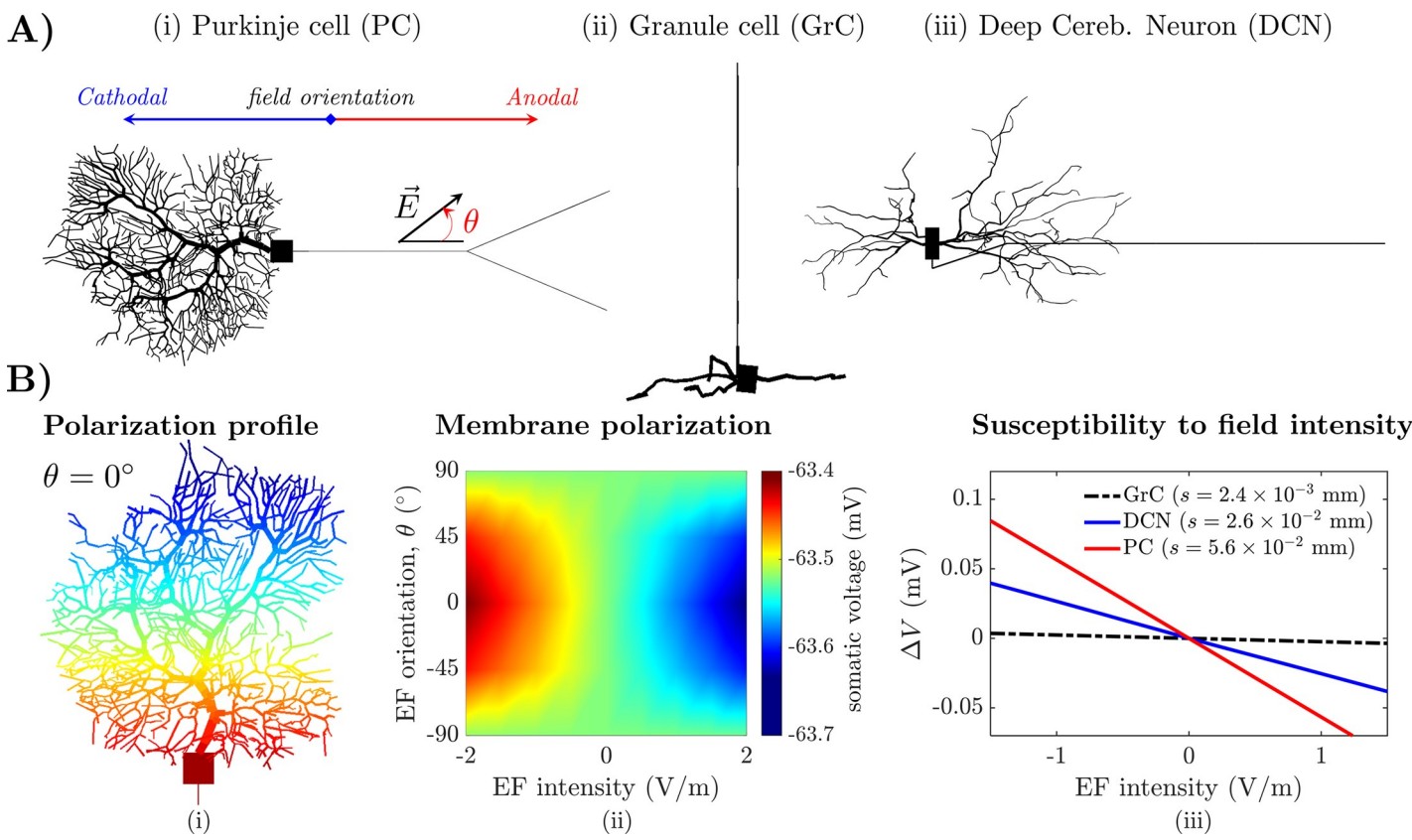

**Fig 3. Coupling EF estimation and multicompartment cerebellar neuron models.** (A) Morphology of the multicompartment neuron models of Purkinje cell (i), granule cell (ii), and deep cerebellar neuron (iii) used in this study. For each model, the orientation of the applied electric field (EF), $\vec{E}$, is measured by the angle, $\theta$, between $\vec{E}$ and the neuron's somato-axonal axis. EF oriented towards the dendrites and the axonal terminals are considered cathodal (*blue arrow*) and anodal (*red arrow*), respectively. (B) Panel (i) shows the polarization profile along the somato-dendritic axis of the PC soma under anodal EF with orientation $\theta = 0°$. Red and blue indicate maximum depolarization and maximum hyperpolarization, respectively (scale not shown). Panel (ii) reports the transmembrane voltage at the PC soma estimated under EF for several combinations of EF intensity and orientation. Negative and positive EF intensities denote anodal and cathodal EF, respectively. Panel (iii) reports the change in somatic transmembrane voltage, $\Delta V$, of the GrC, DCN, and PC model, respectively, as the EF intensity increases ($\theta = 0°$), along with the polarization length ($s$). In (ii)-(iii), neuron models were simulated with blocked sodium channels to prevent the generation of action potentials.

elicited limited variation to the somatic transmembrane voltage at rest (PC: ±0.78 mV; DCN: 1.26 mV; GrC: ±0.12 mV, for EF intensity at ±1.5 V/m, respectively; values calculated while ionic channels are blocked), regardless of the EF orientation or intensity, **Fig 3B (ii)**. The maximum variation, $\Delta V$, to the somatic voltage at rest, instead, occurred when the EF is aligned with the dendritic-somato-axonal axis, i.e., $\theta = \pm 1°$, **Fig 3B (ii)**. The modest variation of the transmembrane voltage stemmed from the small amplitude of the EF-induced polarization across all cell types as well as the small slope, $s$, of the polarization line (a.k.a. "*polarization length*" or "*cell susceptibility*" [13]) as the EF intensity increases, **Fig 3B (iii)**.

The change in resting transmembrane voltage had an impact on the firing rate of the PC and DCN. These neurons exhibit tonic spiking at rest when no tDCS is applied (61.3 Hz versus 27.2 Hz, PC model versus DCN model). For EF intensities up to ±1.5 V/m ($\theta = 0°$), the change in firing rate was noticeable for the PC model (2.14% of the baseline value, i.e., 1.31 Hz) and larger compared to the DCN model (0.29% of the baseline value, i.e., 0.08 Hz), which is consistent with previous *in vitro* findings [43,44]. The GrC model, instead, was silent at rest, and EF intensities up to ±1.5V/m did not facilitate action potentials, likely because of the small geometric size of the GrC model compared to PC and DCN. Small cells like GrC, in fact, have

higher axial resistance and shorter length constants compared to cells like PC and DCN due to smaller diameters (see **S2 Table**), and this determines a lower polarization response to applied electric fields [45].

Altogether, these results indicate that cerebellar neurons have low susceptibility to tDCS currents up to 2 mA. tDCS-induced EF, in fact, would cause generally mild variations to the neurons' baseline polarization and firing pattern at rest. Among these neurons, small cells like GrC remain likely unaffected by tDCS. Purkinje cells and deep cerebellar neurons, instead, have comparable susceptibility to EF but only Purkinje cells will likely experience a change in tonic firing under tDCS.

## 2.3 Rebound firing of deep cerebellar neurons is unaffected by cerebellar tDCS

We then investigated the effect of EF on the response of cerebellar neurons to exogenous stimuli. We focused on GABAergic input from Purkinje cells, as the resultant inhibition can cause a robust rebound spiking in the DCN, whose duration and rate are related to the strength and timing of motor responses [46]. We concurrently applied a single stimulus to all GABAergic synapses disseminated along the DCN dendrites and soma and measured the resultant rebound spiking at the soma, **Fig 4A (i)** (*inset*). The rebound spiking occurred at the end of a rapid repolarization of the DCN membrane (~50 ms), consisted of a sequence of high-frequency action potentials, and terminated with prolonged adaptation of the instantaneous firing rate (IFR), **Fig 4A (i)**.

We found that EF at ±1.5V/m had modest effects on the high-frequency spiking and the subsequent firing adaptation process. The maximum change in IFR was approximately 5 Hz compared to the case without stimulation, **Fig 4A (ii)**, was observed 2-to-3 action potentials after the first rebound spike, and then decayed rapidly to the pre-rebound value within the next few spikes, **Fig 4A (ii)** (*inset*). On average, the maximum change in IFR remained within ±4% of the pre-stimulation value (i.e., firing rate: 98.04 Hz; 99.01 Hz; 101.01 Hz; no tDCS, 1.5 V/m anodal EF, and 1.5 V/m cathodal EF, respectively). At the membrane level, EF polarization produced a modest shift in the chloride reverse potential and a small delay in the activation of the hyperpolarization-activated cyclic nucleotide-gated cationic (HCN) currents, which are the main factors shaping the rebound spiking [35].

Altogether, these results indicate that, even though a similar level of polarization is induced in the deep cerebellar nuclei and the cerebellar cortex, the direct effects of tDCS on glutamatergic deep cerebellar neurons are modest in terms of encoding synaptic stimuli from the Purkinje cells. Accordingly, the influence of tDCS on deep cerebellar nuclei is likely indirect and stems from EF-induced polarization of the presynaptic axon terminals of Purkinje cells. This is expected because the spiking rate of the deep cerebellar neuron depends on the balance between the activation rates of glutamatergic and GABAergic synapses [35]. Moreover, tDCS-induced polarization of presynaptic axon terminals has been shown *in vitro* to alter the efficacy of synaptic transmission [12].

## 2.4 GrC responses to mossy fiber activation are robust against tDCS

Next, we considered the response of the GrC model to applied EF. The GrC model predictions before EF application were compared against *in vitro* data reported in [36]. Specifically, we applied step currents of increasing intensity to the GrC soma and measured the induced firing rate and the latency between the current injection and the time of the first action potential. These two metrics were used to assess the validity of the GrC model. Due to difference in size

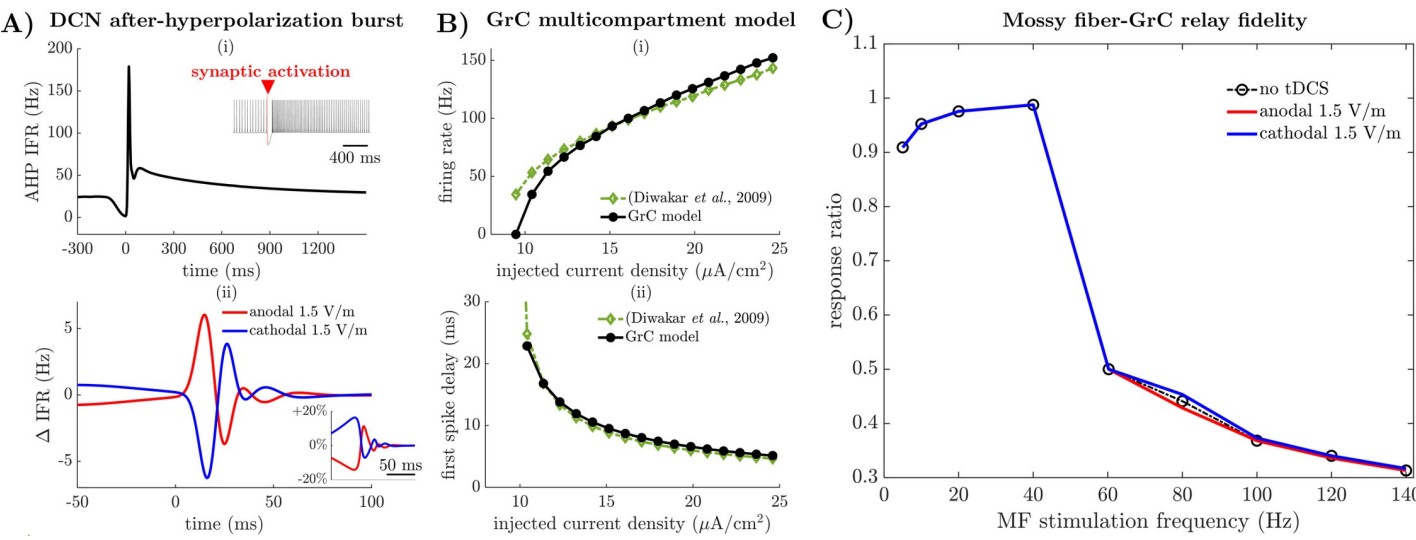

**Fig 4. Predicted effects of EF on deep cerebellar neuron and granule cell models.** (A) Panel (i) shows the instantaneous firing rate (IFR) of the DCN soma versus time during the after-hyperpolarization (AHP) rebound phase when no EF is applied. Time of synaptic activation is $t = 0$ ms. *Inset*: Membrane voltage at the DCN soma during the AHP rebound phase. A red triangle marks the activation of the GABAergic synapse on the DCN dendrites. Panel (ii) shows the variation (Δ) in IFR for the DCN soma during the AHP rebound phase under anodal (*red line*) and cathodal (*blue line*) EF at 1.5V/m. Variation are expressed as differences from the IFR under no EF in (i). *Inset*: Changes are reported as percentual variations from the IFR under no tDCS. (B) Comparison between the multicompartment GrC models proposed in this study (*black lines*) and in [36] (*green lines*). The average somatic firing rate (i) and the first-spike latency (ii) during injection of depolarizing current steps in the GrC soma are shown. (C) Relay fidelity of the GrC soma in response to the tonic activation of presynaptic mossy fibers (MF). The ratio between the number of GrC action potentials and MF stimuli versus the frequency of MF stimuli is depicted when no EF (*black line*) or 1.5 V/m EF (*red line*: anodal; *blue line*: cathodal) is applied.

between the granule cells considered in our study and in [36], the current density (i.e., injected current divided by cell surface) associated with each current step was calculated.

We found that the metrics calculated for the GrC model were closely aligned with the data for step currents at 10 μA/cm$^2$ and above, **Fig 4B (i)-(ii)**. Differently from [36], though, currents less than 8 μA/cm$^2$ did not elicit action potentials, **Fig 4B (i)**. This may be due to the different size of our GrC model, which was reconstructed from a human granule cell and had a larger surface compared to the rodent GrC considered in [36], and the lack of transient receptor potential (TRP) channels [47] in our model. This was further confirmed by the long latency to the first action potential that was reported in [36] for currents around 10 pA (i.e., 10 μA/cm$^2$), **Fig 4B (ii)**.

The GrC model was then used to investigate the fidelity of the GrC response to mossy fibers (MF) under EF. Recent evidence indicate that granule cells can show adaptation and acceleration in response to MF as the frequency of MF stimuli increase, and such adaptation can directly affect short-term synaptic plasticity throughout the cerebellar cortex [47].

We applied glutamatergic synapses on the GrC dendrites and mimicked MF activation with a regular sequence of presynaptic stimuli whose frequency varied linearly up to 140 Hz. For every value of the MF stimulation frequency, the GrC model was allowed to reach steady state, and then the relay fidelity of the GrC was measured as the ratio, $r$, between the number of action potentials evoked at the soma and the number of MF stimuli delivered in a 2,000-ms window.

We found that, under no tDCS, adaptation emerged for MF stimulation frequencies greater than 40 Hz and resulted in a rapid reduction of the relay fidelity $r$, followed by a linear decay for frequencies up to 140 Hz, **Fig 4C**. Compared to no tDCS, -1.5 V/m EF (anodal) and +1.5 V/m EF (cathodal) induced mild changes to the relay fidelity. These changes depended on the MF stimulation frequency and were mainly observed for 80 Hz MF stimulation, **Fig 4C**. In

this case, anodal EF decreased the relay fidelity ratio ($r = 0.429$; 2.81% decrease) compared to the baseline value under no EF ($r = 0.441$). Cathodal EF, instead, increased the ratio ($r = 0.453$, i.e., 2.81% increase). For MF stimulation frequencies beyond 80 Hz, the difference between the values of $r$ under anodal EF and cathodal EF was less than 1.2% of the baseline value at all times and for every EF intensity up to 1.5 V/m.

Overall, these results suggest that tDCS-induced EF have limited effect on the firing activity of granule cells. Combined with the modest modulation of DCN reported above, these results suggest that the modulatory effect of cerebellar tDCS primarily focuses on the outer layers, with Purkinje cells in the cortex being the most prominent target.

## 2.5 Cerebellar tDCS modulates the response of Purkinje cells to synaptic stimuli

We quantified the effects of cerebellar tDCS on the response of Purkinje cells to synaptic stimuli. First, we measured the transmission delay, $T_d$, which is the time required for an impulse at the most distal dendritic compartment to travel to the soma and evoke an action potential. As in [34], we blocked $Na^+$ and $Ca^{2+}$ channels in the axonal initial segment of the PC model and let the soma compartment transitioning from silence to tonic spiking in response to a distal dendritic input, **Fig 5A** (*inset*).

We found that, while the steady-state firing rate (FR) at the soma was modestly affected and varied linearly with the EF intensity (correlation value $R = 0.99$; coefficient of determination $R^2 = 0.97$), the transmission delay was dramatically affected by EF orientation and intensity. First, anodal EF consistently reduced $T_d$ while cathodal EF significantly increased $T_d$, **Fig 5A**. Second, the percentual variation of $T_d$ from the baseline value (i.e., $T_d = 8.5$ ms when no EF was applied) increased linearly with the EF intensity (**Fig 5A**), ranged from a 71.8% decrease ($T_d = 2.4$ ms, 1.5 V/m anodal EF) to a 134.1% increase ($T_d = 19.9$ ms, 1.5 V/m cathodal EF), and resulted in an average variation for $T_d$ of 5.8 ms per 1 V/m increment of the electric field.

Because of the importance of PC spike timing in learning and coordination [48,49], we also measured the effects of EF on the phase response curve (PRC) of the Purkinje cells when parallel fibers are activated. The PRC measures the sensitivity of Purkinje cells to the presynaptic stimuli [50], and therefore changes to the PRC indicate a shift in the excitability of Purkinje cells. **Fig 5B** shows the PRC estimated when no tDCS is applied and for EF at ±1.5 V/m. The effects of the EF were mainly concentrated at phase $\phi = 0°$, where anodal EF produced a maximum anticipation of the phase response by 6.2%, which corresponds to a 0.66-ms-reduction in latency between the synaptic stimulus and the somatic action potential. Cathodal EF, instead, resulted in a maximum phase increment of 5.3%, i.e., 0.65 ms increase in latency. No significant alteration was reported, instead, at higher phases for anodal or cathodal EF, while changes to the PRC at low phase values remained less than 5% for EF below 1.5 V/m.

Altogether, these results suggest that tDCS has a modest impact on the sensitivity of Purkinje cells to inputs from the parallel fibers. The impact can be attributed primarily to the response of Purkinje cells to stimuli arriving close to simple spikes and results in a modulation of less than 1 ms. Hence, although it is possible that tDCS alters the cerebellar response to cortical projections, the effects of cerebellar tDCS are likely focused on the activity along the parallel fibers and/or the potentiation of the synapses onto the Purkinje cells rather than on the encoding capabilities of Purkinje cells in response to cortical projections.

## 2.6 Cerebellar tDCS modulates the synaptic inhibition of Purkinje cells

We then considered the effects of tDCS on the spiking pattern that Purkinje cells display in response to input from the parallel fibers. We created trains of presynaptic stimuli for each

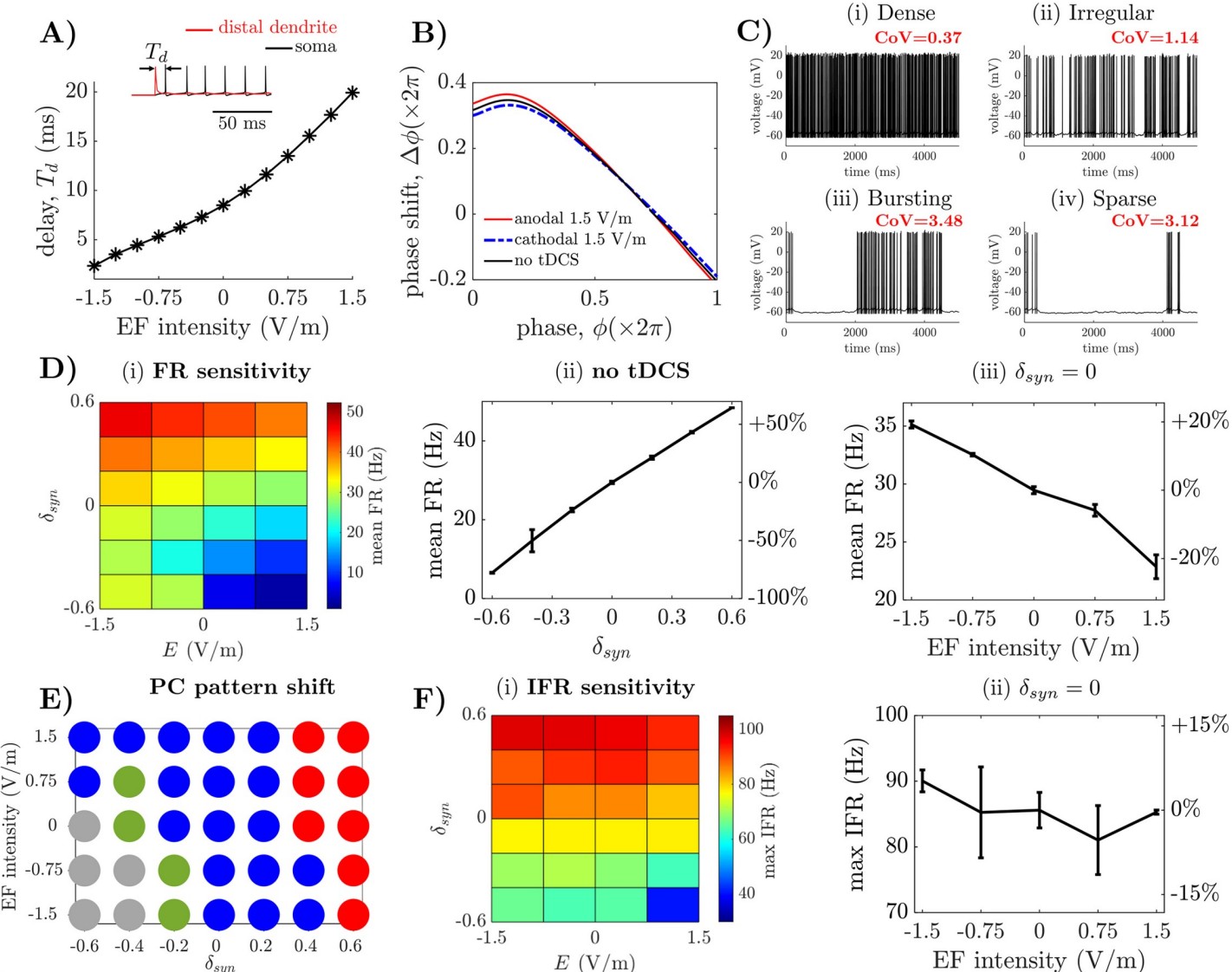

**Fig 5. Predicted effects of EF on Purkinje cell models.** (A) Transmission delay, $T_d$, of the PC in response to distal dendritic stimuli under EF versus EF intensity. *Inset*: Definition of $T_d$ as interval between the post-stimulus activation of the dendrite (*red*) and the action potential at the soma (*black*). (B) Phase-response curve (PRC) of the PC model estimated without stimulation (*black line*) and EF at 1.5 V/m (*red line*: anodal; *blue line*: cathodal). (C) Examples of transmembrane voltage at the PC soma during dense (i), irregular (ii), bursting (iii) and sparse (iv) firing, respectively, with no EF. For each example, the coefficient of variation (CoV) of the spiking pattern is reported. (D) Effects of EF on the postsynaptic firing rate (FR) at the PC soma. Panels (i), (ii), (iii) show the mean FR at the soma for different combinations of $\delta_{syn}$ and EF intensity, $E$ (i), mean FR versus $\delta_{syn}$ ($E$ = 0 V/m) (ii), and mean FR versus EF intensity ($\delta_{syn}$ = 0) (iii), respectively. (E) Combinations of EF intensity and $\delta_{syn}$ for which the PC soma exhibits dense (*red*), irregular (*blue*), bursting (*green*), or sparse (*gray*) firing. (F) Effects of EF on the postsynaptic instantaneous firing rate (IFR). Panels (i)-(ii) show the maximum IFR at the soma for various combinations of $\delta_{syn}$ and EF intensity, $E$ (i), and the maximum IFR vs. EF intensity ($\delta_{syn}$ = 0) (ii), respectively.

one of over 1,000 glutamatergic and GABAergic synapses on the PC dendrites, and we studied the resultant spiking pattern elicited at the PC soma, both with and without EF application. The trains of presynaptic stimuli were designed to reproduce the combination of feedforward inhibition due to granular layer interneurons and diffused depolarization due to parallel fibers [51,52], and resulted in a cumulative presynaptic input to the PC model with spectral

proprieties similar to those of field potentials recorded *in vivo* at rest in rodent models [53]. See **S3A Fig** for a power spectrum analysis of the cumulative synaptic activation sequence.

Depending on the synaptic gain scaling factor, $\delta_{syn}$, used in the definition of the synaptic input (see definition in the Methods section), four types of discharge patterns were defined at the soma compartment when no stimulation is applied, i.e., *dense* (inter-spike intervals [ISI] are less than 80 ms, and the ISI coefficient of variance [CoV] is less than 0.5), *irregular* (the maximum ISI value is between 80 ms and 200 ms, and the ISI CoV value is between 0.5 and 2), *bursting* (the maximum ISI value is above 200 ms, and the ISI CoV value is above 2), or *sparse* (i.e., the pattern includes only a few spikes followed by a silent periods exceeding 3,000 ms, and the ISI CoV, if defined, is greater than 3), see example patterns in **Fig 5C (i)-(iv)**, respectively. Conditions for firing pattern classification were modified from [54], and the classification of *bursting* patterns versus *sparse* patterns was set based on inspection of the cases with CoV>3.

In case of no stimulation, the average somatic firing rate (FR) across all discharge patterns was 29.5 ± 0.3 Hz ($\delta_{syn}$ = 0) and ranged between 6.5 ± 0.1 Hz ($\delta_{syn}$ = -0.6) and 48.4 ± 0.1 Hz ($\delta_{syn}$ = +0.6), **Fig 5D (i)** (mean ± S.D. across three distinct combinations of presynaptic trains and synapse locations on the PC dendrites). The range of FR was consistent with the firing rate values observed *in vivo* (e.g., see Table 1 in [55]) and depended linearly on $\delta_{syn}$, **Fig 5D (ii)** (slope = 173.2 Hz per unit increment of $\delta_{syn}$, $R^2$ = 0.966).

Applied EF resulted in two significant changes to the PC firing pattern. First, the average FR at the PC soma varied widely in response to EF intensity, **Fig 5D (iii)** and **Table 1**. For $\delta_{syn}$ >0, EF at 1.5 V/m and EF at 0.75 V/m varied the FR up to 25% and 12%, respectively, compared to the baseline FR (i.e., estimated when no EF is applied), **Fig 5D (i)**, and the variation was statistically significant compared to baseline (two-sample *t*-test, *P*-value *P*<0.01, **Table 1**). For reductions in the synaptic conductance exceeding 20% (i.e., $\delta_{syn}$<-0.2), cathodal EF

**Table 1. Effects of tDCS on Average PC Firing Rate.** Effects of the tDCS-induced EF intensity, *E*, and the synaptic gain on the average firing rate at the PC soma. $\delta_{syn}$ indicates the variation from the baseline synaptic gain expressed in percentage. Firing rates are in Hz, and percentual changes in firing rate are computed with respect to the no-tDCS case under the same level of synaptic gain.

| | | E | | | | |
|---|---|---|---|---|---|---|
| | | -1.5 V/m | -0.75 V/m | 0 V/m | +0.75 V/m | +1.5 V/m |
| $\delta_{syn}$ | -60% | 31.1±1.6 ** | 29.2±0.7 ** | 6.5±0.1 | 3.0±2.8 | 1.0±0.0 ** |
| | | (+376.5%) | (+346.9%) | | (-54.1%) | (-84.7%) |
| | -40% | 29.3±1.1 ** | 21.5±0.4 ** | 14.7±2.8 | 9.8±2.6 * | 2.7±0.4 ** |
| | | (+100.0%) | (+46.4%) | | (-33.2%) | (-81.8%) |
| | -20% | 31.1±0.4 ** | 27.9±0.3 ** | 22.5±0.5 | 18.4±0.2 ** | 13.3±0.8 ** |
| | | (+38.6%) | (+24.3%) | | (-18.1%) | (-40.9%) |
| | 0% | 35.1±0.3 ** | 32.5±0.1 ** | 29.5±0.3 | 27.7±0.5 ** | 22.9±1.0 ** |
| | | (+19.2%) | (+10.4%) | | (-5.9%) | (-22.4%) |
| | 20% | 40.2±0.2 ** | 37.7±0.1 ** | 35.7±0.4 | 33.3±0.3 ** | 29.9±0.3 ** |
| | | (+12.5%) | (+5.4%) | | (-6.7%) | (-16.2%) |
| | 40% | 46.7±0.3 ** | 43.9±0.3 ** | 42.2±0.2 | 39.8±0.2 ** | 37.3±0.2 ** |
| | | (+10.7%) | (+3.9%) | | (-5.7%) | (-11.7%) |
| | 60% | 52.5±0.1 ** | 50.3±0.1 ** | 48.4±0.0 | 46.3±0.3 ** | 44.1±0.1 ** |
| | | (+8.4%) | (+4.0%) | | (-4.3%) | (-8.8%) |

Significance level

*$P$<0.05

** $P$<0.01 under two-sample *t*-test.

reduced the average FR by up to 84.7% compared to baseline (EF intensity: 1.5V/m; $\delta_{syn}$ = -0.6, two-sample $t$-test, $P$-value $P<0.01$), while anodal EF maintained the baseline FR (31.1 ± 1.6 Hz; EF intensity: -1.5V/m; $\delta_{syn}$ = -0.6; no significant difference from the baseline 29.5 ± 0.3 Hz; two-sample $t$-test).

In addition, EF effectively modulated the spiking pattern at the PC soma. First, the CoV of the ISI was highly sensitive to the combination of the synaptic gain and EF intensity, with the largest variations occurring for $\delta_{syn}<0$, **S3B Fig**. Secondly, although the type of pattern remained largely consistent as the EF intensity varied, we found that the PC soma can transition from an irregular pattern to a dense, bursting, or sparse pattern for EF intensities as low as 0.75 V/m, depending on the value of $\delta_{syn}$, and the shift in pattern is more likely to occur for $\delta_{syn}<0$, **Fig 5E**.

Finally, to assess whether these changes in pattern could be due to an alteration of the baseline FR of the PC model before synaptic activation, we measured the maximum instantaneous firing rate (IFR) at the soma for several combinations of synaptic gain $\delta_{syn}$ and EF intensity. We found that, while the maximum IFR was highly dependent of the synaptic gain, i.e., it varied from 64.4 Hz to 98.9 Hz as $\delta_{syn}$ ranged from -0.6 to +0.6, **Fig 5F (i)**, the maximum IFR was modestly affected by the EF intensity, see **Fig 5F (ii)** for the case $\delta_{syn}$ = 0 and **Table 2** for other values of $\delta_{syn}$.

The behavior of the IFR under EF indicates that the variation of PC discharge pattern did not result from a modulation of the PC spontaneous FR, i.e., before synaptic activation. Instead, it likely depends on the polarization of the cell membrane, which is either shifted towards values closer to the spiking threshold (i.e., in case of anodal EF) or towards values further away from the spiking threshold (i.e., in case of cathodal EF). To further clarify this point, we considered the scenario reported in **S3C Fig**. Here, we set $\delta_{syn}$ = -0.4, used the presynaptic pulse trains in **Fig 5D–5F**, and varied the EF intensity between -1.5V/m and +1.5V/m. Panels

**Table 2. Effects of tDCS on Maximum Instantaneous PC Firing Rate.** Effects of the tDCS-induced EF intensity, $E$, and the synaptic gain on the maximum instantaneous firing rate (IFR) at the PC soma. $\delta_{syn}$ indicates the change from the baseline synaptic gain expressed in percentage. IFR values are in Hz, and percentual changes in firing rate are computed with respect to the no-tDCS case under the same level of synaptic gain.

| | | $E$ | | | | |
|---|---|---|---|---|---|---|
| | | **-1.5 V/m** | **-0.75 V/m** | **0 V/m** | **+0.75 V/m** | **+1.5 V/m** |
| $\delta_{syn}$ | **-60%** | 65.7±1.5 | 63.6±0.7 | 64.4±3.9 | 40.4±11.4 [*] | 29.7±0.7 [**] |
| | | (+2.0%) | (-1.3%) | | (-37.2%) | (-53.9%) |
| | **-40%** | 69.6±1.5 | 71.2±1.7 | 68.2±5.6 | 62.8±2.7 | 57.6±2.3 |
| | | (+2.0%) | (+4.4%) | | (-7.9%) | (-15.5%) |
| | **-20%** | 77.3±2.9 | 77.7±1.9 | 77.5±4.6 | 78.0±1.8 | 66.1±3.4 |
| | | (-0.2%) | (+0.3%) | | (+0.7%) | (-14.6%) |
| | **0%** | 90.0±1.7 | 85.3±6.9 | 85.6±2.7 | 81.0±5.2 | 85.3±0.3 |
| | | -5.20% | (-0.4%) | | (-5.3%) | (-0.4%) |
| | **20%** | 89.2±1.5 | 92.0±0.9 | 93.4±2.8 | 89.4±4.5 | 88.0±3.5 |
| | | (-4.6%) | (-1.5%) | | (-4.3%) | (-5.8%) |
| | **40%** | 97.9±1.8 | 97.7±1.3 | 96.6±1.5 | 92.8±1.4 | 93.3±3.9 |
| | | (+1.3%) | -1.10% | | (-3.9%) | (-3.4%) |
| | **60%** | 105.1±3.0 | 102.8±4.8 | 98.9±3.7 | 101.5±6.4 | 96.9±2.1 |
| | | (+6.4%) | (+4.0%) | | (+2.6%) | (-2.0%) |

Significance level

[*] $P<0.05$

[**] $P<0.01$ under two-sample $t$-test.

(i)-(v) in **S3C Fig** report the synaptic current $I_{syn}(t)$ (*red line*, see definition in Eq (2) in the Methods section) after low-pass filtering (5[th] order Butterworth filter, cutoff frequency: 30 Hz), and the somatic transmembrane voltage (*black line*) for different EF intensities. We observed that, while EF affected $I_{syn}(t)$ moderately, the resultant spiking pattern varied significantly and switched from bursting without stimulation, **S3C (iii) Fig**, to irregular under anodic EF, **S3C (i) and (ii) Fig**, or drastically thinned under cathodal EF, **S3C (iv) and (v) Fig**. Under cathodal EF, the inhibitory effects of the hyperpolarizing synaptic currents were amplified and resulted in a prolonged suppression of spiking.

Altogether, these results indicate that, through modulation of the postsynaptic activation, cerebellar tDCS can alter the timing and duration of the feedforward inhibition to the Purkinje cells. This would result in a modulation of the spiking pattern elicited by bouts of excitatory synaptic stimuli, but the magnitude of such modulation critically depends on the strength of the synapses along the PC dendritic tree. The latter point is interesting because previous studies [43,44] have primarily used alternate, high-intensity EF to modulate the firing pattern of Purkinje cells and have shown that Purkinje cells can be entrained in a common, EF-driven rhythm. Our simulations now suggest that constant, low-intensity EF can also modulate Purkinje cells, but the mechanisms of modulation would be different compared to alternate EF. While no common rhythm would be evoked, constant low-intensity EF can steer Purkinje cells to discharge patterns that have similar statistical properties. Also, given the range of EF intensities reported across the cerebellar region, this similarity can be promoted on neurons across a large region of cerebellar cortex, thus suggesting that the effects of tDCS could be diffused well beyond the area under the electrode.

Finally, these results suggest that anodal stimulation can impede the emergence of bursting patterns in favor of an irregular, upregulated spiking activity by lifting the PC above the spiking threshold, whereas cathodal EFs likely result in a more sporadic spiking due to membrane hyperpolarization, which may be associated with reduced neural encoding and motor learning [56].

## 2.7 Cerebellar tDCS modulates the silence period following complex spikes

A complex spike (CS) is generated at the soma of Purkinje cells in response to climbing fiber activation and consists of a large spike followed by 2 to 4 spikelets [57,58], **Fig 6A (i)**. Complex spikes are important to the cerebellar function as they mediate the response to unconditioned stimuli and control the redistribution of the synaptic weights across different dendritic branches of the Purkinje cell [59,60].

We found that the morphology and duration of the CS were affected by tDCS, where changes were modest in size compared to the no stimulation case, and no trend occurred as the EF orientation and intensity changed, **Fig 6 (i)-(iii)**, even though cathodal EFs occasionally resulted in one additional spikelet at the end of the CS, e.g., **Fig 6A (iii)**.

The silence period, $T_{sp}$, between a CS and the following simple spike (**Fig 6B**), instead, is an important feature of complex spikes [61] and was significantly affected by tDCS. Specifically, **Fig 6C** reports $T_{sp}$ as a function of the EF intensity and indicates that anodal EF caused consistent reductions (i.e., up to -22.4% compared to no tDCS) in silence period while cathodal EF caused a remarkable prolongation of the silence period, i.e., up to +80% of the value under no tDCS ($T_{sp}$ = 440.6 ms, 342.1 ms, and 792.4 ms for no EF, -1.5 V/m EF, and +1.5 V/m EF, respectively).

To further understand the cellular mechanisms that mediate the variation in silence period, we considered the PC model under no EF and selectively applied the EF-induced polarization to the soma compartment, while the other compartments remained unaltered. This resulted in a bias, $V_{bias}$, to the transmembrane voltage at the soma versus the other compartments, with

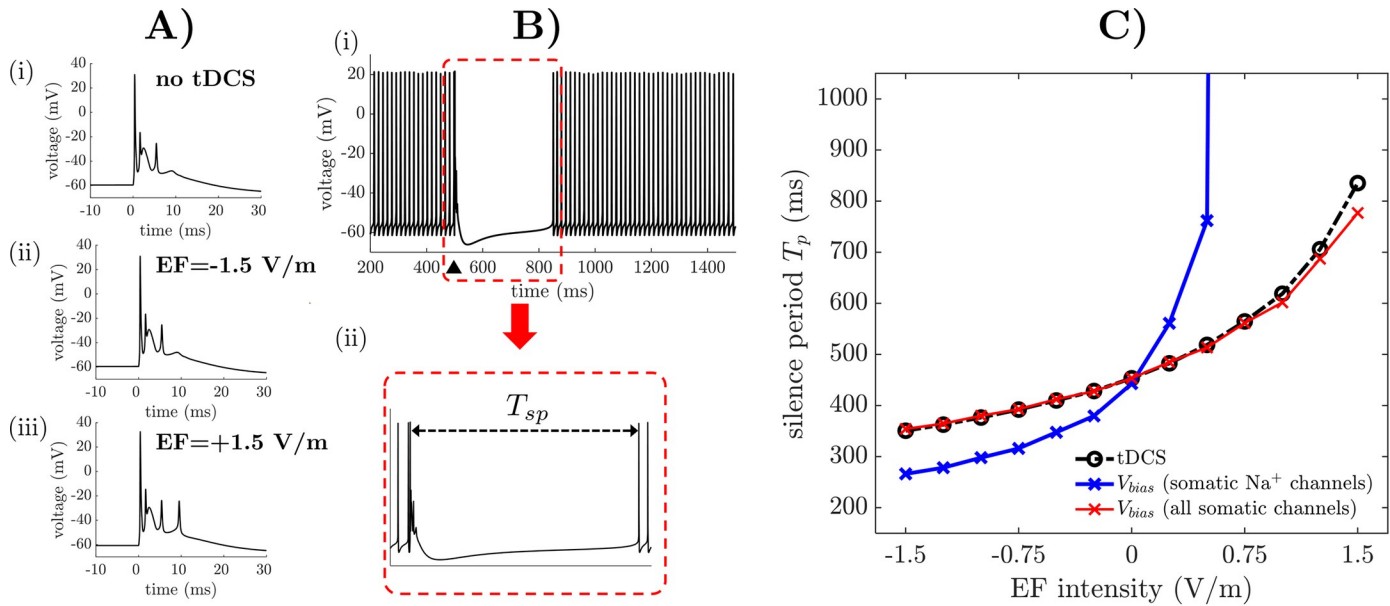

**Fig 6. Predicted effects of EF on PC complex spiking and silence period.** (A) Examples of complex spike (CS) generated by the PC model when no EF is applied (i) and EF with intensity -1.5 V/m (ii) and +1.5 V/m (iii) is applied, respectively. (B) Panel (i) shows that evoking a CS (*black triangle*) disrupts the tonic spiking of the PC soma and elicits a prolonged repolarization phase. Panel (ii) magnifies the area within the dashed red box in (i) and defines the silence period $T_{sp}$ as the interval between the highest spikelet of the CS and the first spike after the repolarization phase following the CS. (C) $T_{sp}$ versus EF intensity (*black line*). $T_{sp}$ versus EF intensity is also reported when no EF is applied and a bias, $V_{bias}$, is applied to the Nav1.6 channels only (*blue line*) or all ionic channels (*red line*) of the PC soma. For every field intensity, $V_{bias}$ is the polarization induced by the EF on the ionic channels (range: ±0.215 mV).

$V_{bias}$ depending on the soma's length constant (range: ±0.215 mV for EF at ±1.5V/m, see *red line* in **Fig 6C**).

Alternatively, we considered the PC model under no stimulation and selectively applied the bias $V_{bias}$ to the transmembrane voltage applied to somatic Na$^+$ v1.6 (Nav1.6) channels (**Fig 6C**, *blue line*). Remaining ionic channels and compartments, instead, were unaltered. Nav1.6 channels are responsible for the initiation of simple spikes and mediate the rapid recovery from inactivation in Purkinje cells, thus controlling the duration of the silent period [62]. Accordingly, this manipulation aimed to assess whether a shift in the Nav1.6 dynamics due to the polarization of the cell membrane can modulate the recovery period leading to simple spiking.

We found that a selective polarization of the soma resulted in a modulation of the silence period like the nominal case (**Fig 6C**, *red line*). Vice versa, a selective alteration of the Nav1.6 channel dynamics resulted in a more rapid variation of the silence period (**Fig 6C**, *blue line*), with higher reduction in $T_{sp}$ for negative values of $V_{bias}$, which correspond to anodal stimulation, and a prolongation of the silence period beyond 10,000 ms for $V_{bias} \geq 0.1$ mV.

Altogether, these results indicate that the modulation of the silence period under tDCS is primarily mediated by somatic polarization. A change in transmembrane voltage at the soma likely causes a rapid modulation of the Nav1.6 channels, which is partially compensated by the activation of K$^+$ channels. This would result in higher sensitivity of the CS to cathodal EF versus anodal EF. This is further confirmed by the fact that Nav1.6-mediated currents are highly sensitive to EF intensity compared to the remaining currents, see **S4 Table**.

## 2.8 Cerebellar tDCS modulates spontaneous bursting in Purkinje cells

Purkinje cells exhibit a bursting pattern with limited duration upon receiving steady depolarization [63], e.g., see **Fig 7A** for a 2-nA-step current injected in the soma at time $t = 200$ ms.

When observed *in vivo*, this bursting can contribute to the short-term memory capabilities of the cerebellar network [64,65] and is mediated by a sustained depolarization of the PC dendrites secondary to the activation of granule cell axons [23]. It can also be elicited by activating climbing fibers in response to sensory stimuli [66].

The effects of applied EF on the burst period, $T_{burst}$ (**Fig 4A**, *inset*), and the count of bursts in the entire sequence were assessed as the EF polarization and intensity varied. We found that the duration of the transient bursting sequence and the burst count varied up to 25% for EF at ±1.5 V/m (duration burst sequence: 1,620 ms, 1,207 ms, and 1,993 ms; burst count = 12, 9, and 15; values reported for no EF, -1.5 V/m EF, and +1.5 V/m EF, respectively). The burst period, instead, increased over consecutive bursts as the transient bursting pattern decreases, **Fig 7B**.

By further decomposing the burst period into an inter-burst interval and a spiking phase ($T_{intv}$ and $T_{spike}$, respectively, **Fig 7A** *inset*), we found that the variation in the burst period was mainly contributed by the inter-burst interval, whose relationship with the number of bursts follows a similar trend as $T_{burst}$, **Fig 7C**. In contrast, the spiking period, $T_{spike}$, of each burst varied modestly under EF, with no significant trend across the EF intensities, **Fig 7D**. Furthermore, the average value, $\overline{T}_{burst}$, of the burst period over the first five consecutive bursts was 4% higher under anodal EF and 3% lower under cathodal EF compared to baseline (no EF: 116.7 ± 6.0 ms; -1.5 V/m EF: 121.1 ± 7.6 ms; +1.5 V/m EF: 113.4 ± 5.0 ms, respectively; mean ± S.D.).

To further understand the cellular mechanisms that mediate the rate adaptation during transient bursting, we considered the PC model under no EF and selectively applied the polarization due to the applied EF to the soma, while the other compartments remained unaltered (*case 1*), or the dendrites, while the other compartments remained unaltered (*case 2*). The reason for comparing *case 1* versus *case 2* is that the transient bursting pattern may be regulated by calcium ($Ca^{2+}$) channels distributed across the dendrites [25].

We selectively applied a bias, $V_{bias}$, to the transmembrane voltage of the compartments of interest in both cases, and we determined the resultant burst count (**Fig 7E**) and mean burst period (**Fig 7F**). We found that the dendritic polarization (*case 2*, green lines in **Fig 7E and 7F**) under applied EF modestly altered the mean burst period (e.g., $T_{burst}$ slightly decreases for EF above 0.6 V/m) with no change to the burst count, while the adaptation of the firing pattern over consecutive bursts was mainly mediated by somatic polarization (*case 1*). Under somatic polarization, in fact, the burst count ranged from 9 to 15 (**Fig 7E**, *red squares*) and the mean burst period ranged from 113.8 ± 5.9 ms to 120.9 ± 7.3 ms (**Fig 7F**, *red line*), which correspond to a variation of -2.5% and +3.7% from the value under no EF, respectively.

Finally, we determined whether the burst adaptation depends on the polarization of all somatic ionic channels or mainly the voltage-gated $K^+$ channels (Kv1.1, Kv3.4, and Kir2.3, [34]), as these channels are expected to control the inter-burst interval [25]. We found that a selective polarization of the voltage-gated $K^+$ channels resulted in a similar number of bursts as in *case 1* (i.e., polarization of all somatic ionic channels), **Fig 7E** (*blue circles*) but led to an oversensitivity of the mean burst period to the EF intensity (**Fig 7F**, *blue line*, percentual change from the no-stimulation case: -8.4% to +11.0%). Vice versa, a selective polarization of somatic $Na^+$ and $Ca^{2+}$ channels or calcium-activated $K^+$ channels (rather than the voltage-gated $K^+$ channels) resulted in negligible changes to the burst count and mean burst period, i.e., less than 1%.

Altogether, these results indicate that tDCS-induced EF can modulate transient bursts and rate adaptation in Purkinje cells by up to 25% of the baseline value. This modulation is primarily mediated by the polarization of the somatic ionic channels, with the changes to the burst period being primarily driven by modulation of the $K^+$ currents.

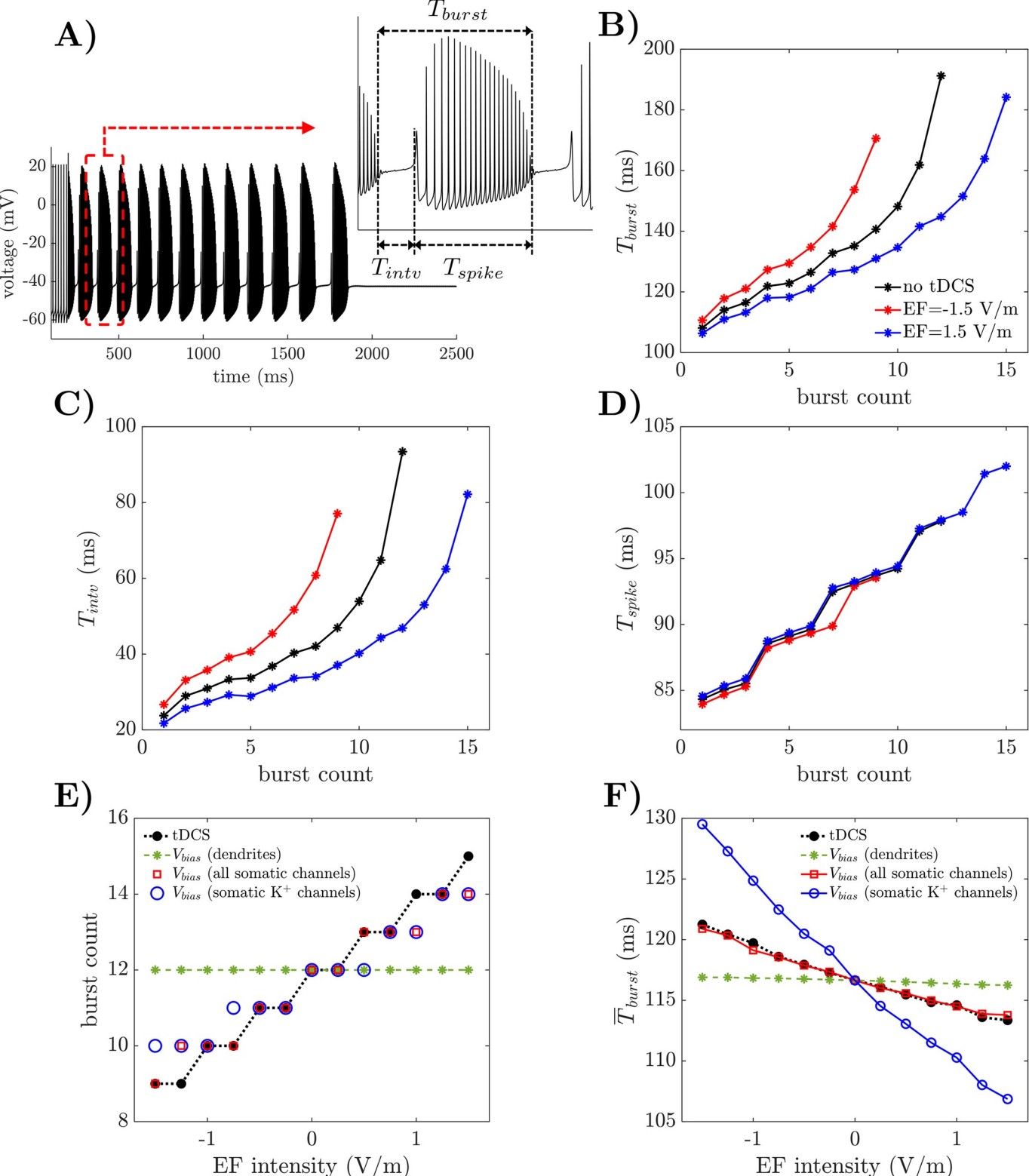

**Fig 7. Predicted effects of EF on PC bursting.** (A) Spontaneous bursting at the PC soma. The somatic transmembrane voltage is depicted when no EF is applied. *Inset*: Definition of burst period ($T_{burst}$), burst interval ($T_{intv}$), and spiking period ($T_{spike}$). (B-D) $T_{burst}$ (B), $T_{intv}$ (C), and $T_{spike}$ (D) are reported for consecutive bursts at the PC soma under no EF (*black line*) and 1.5 V/m EF (*red line*: anodal; *blue line*: cathodal), respectively. (E) Count of consecutive bursts at the PC soma versus EF intensity (*black line*). Burst count versus EF intensity is also reported when no EF is applied and a bias, $V_{bias}$, is applied to all dendritic ionic channels

(*green line*), all somatic ionic channels (*red squares*), or somatic K$^+$ channels only (i.e., Kv1.1, Kv3.4, and Kir2.3; *blue circles*). (F) Mean burst period, $\overline{T}_{burst}$, at the soma versus EF intensity (*black line*). $\overline{T}_{burst}$ versus EF intensity is also reported when no EF is applied and a bias, $V_{bias}$, is applied to all dendritic ionic channels (*green line*), all somatic ionic channels (*red line*), or somatic K$^+$ channels only (*blue line*). $\overline{T}_{burst}$ is the average value of $T_{burst}$ across the first 5 consecutive bursts. For every EF intensity in (E-F), $V_{bias}$ is the polarization induced by EF on the ionic channels (range: ±0.215 mV).

## 2.9 Spatial distribution of the PC response to cerebellar tDCS

We considered the EF induced by R-tDCS with pad electrodes (**Fig 1A**) on each subject and determined the distribution of the angle, $\theta$ (**Fig 2A**), between EF and the somatic-axonal axis of the Purkinje cells along the cerebellar cortex. We then used $\theta$ to estimate the polarization induced on the Purkinje cells in every voxel along the cerebellar cortex. We repeated this for the EF induced by anodal tDCS and cathodal tDCS. The procedure to compute the EF projections for the segmented cerebellar cortices is reported in the Methods section.

**Fig 8A** reports the intensity of the EF projections across all voxels covering the cerebellar cortex of atlas 5 (both hemispheres). In case of anodal tDCS, EF projections with high intensity were mainly concentrated on the dorsal side of the cerebellum and on the left hemisphere, while, in case of cathodal tDCS, EF projections with high intensity were mainly concentrated on the rostral side of the cerebellum. PC dendritic trees, which are mainly located along the convex gyri of the cerebellar cortex, received anodal polarization (i.e., negative values and warm colors on the color scale). PC axonal terminals, instead, are primarily located along the concave sulci and were aligned with the entering EF, thus receiving mainly cathodal polarization (i.e., positive values and cool colors on the color scale). The intensity of the EF projections had a similar distribution across hemispheres for the remaining subjects, see **S4A Fig**.

Across all subjects, the portion of left cerebellar cortex where Purkinje cells receive a projected EF higher than 0.5V/m accrued to 3.0 ± 0.6% for anodal tDCS (1.16 ± 0.53% received EF <-1.5 V/m) and 0.93 ± 0.37% for cathodal tDCS (0.47 ± 0.48% received EF > 1.5V/m). The portion of right cerebellar cortex that received a projected EF higher than 0.5 V/m was 1.56 ± 1.12% (anodal tDCS) and 0.37 ± 0.23% (tDCS). **Table 3** reports the cumulative percentage of cerebellar cortex across both hemispheres for every subject and averaged across all subjects. Smaller percentages, instead, were obtained with HD-tDCS configurations because of the reduced spatial extent of the electric field induced by tDCS, see **S4B (i)-(ii) Fig** for HD-tDCS with two and five ring electrodes, respectively.

Altogether, these results indicate that anodal tDCS evokes a net polarization of the cerebellum on both hemispheres, even though the largest polarization occurs in the hemisphere underneath the electrode pad. Moreover, **Fig 8B** reports the distribution of the angle $\theta$ in both hemispheres and indicates that the distribution exhibited symmetry around $\theta = 90°$, was dispersed across the entire range 0°-180° (86.8% of the distribution falls in the range 30°-150°) in the targeted (left) hemisphere and had a unimodal distribution in the right hemisphere. A validation of the method used to estimate the angle $\theta$ between the PC axis and the EF is shown in **S5 Fig** (details in **S1 Text**).

The EF projections also showed a symmetric distribution in both hemispheres, with almost equal populations of Purkinje cells receiving polarization effects that are consistent with anodal stimulation and cathodal stimulation, **Fig 8C**. A significant difference, though, occurred between hemispheres (PC population that received anodal EF projections: 52.0 ± 0.31% versus 48.1% ± 0.25%, left versus right hemisphere; two-sample *t*-test, P-value $P<0.001$). We estimated that, across the whole cerebellum, roughly equal numbers of Purkinje cells are either depolarized by anodal EF or hyperpolarized by cathodal EF, even though there is a slight preference over one direction depending on the electrode polarity over the cerebellum. Specifically, when the anodal electrode is placed on the cerebellum, a few more PCs would be

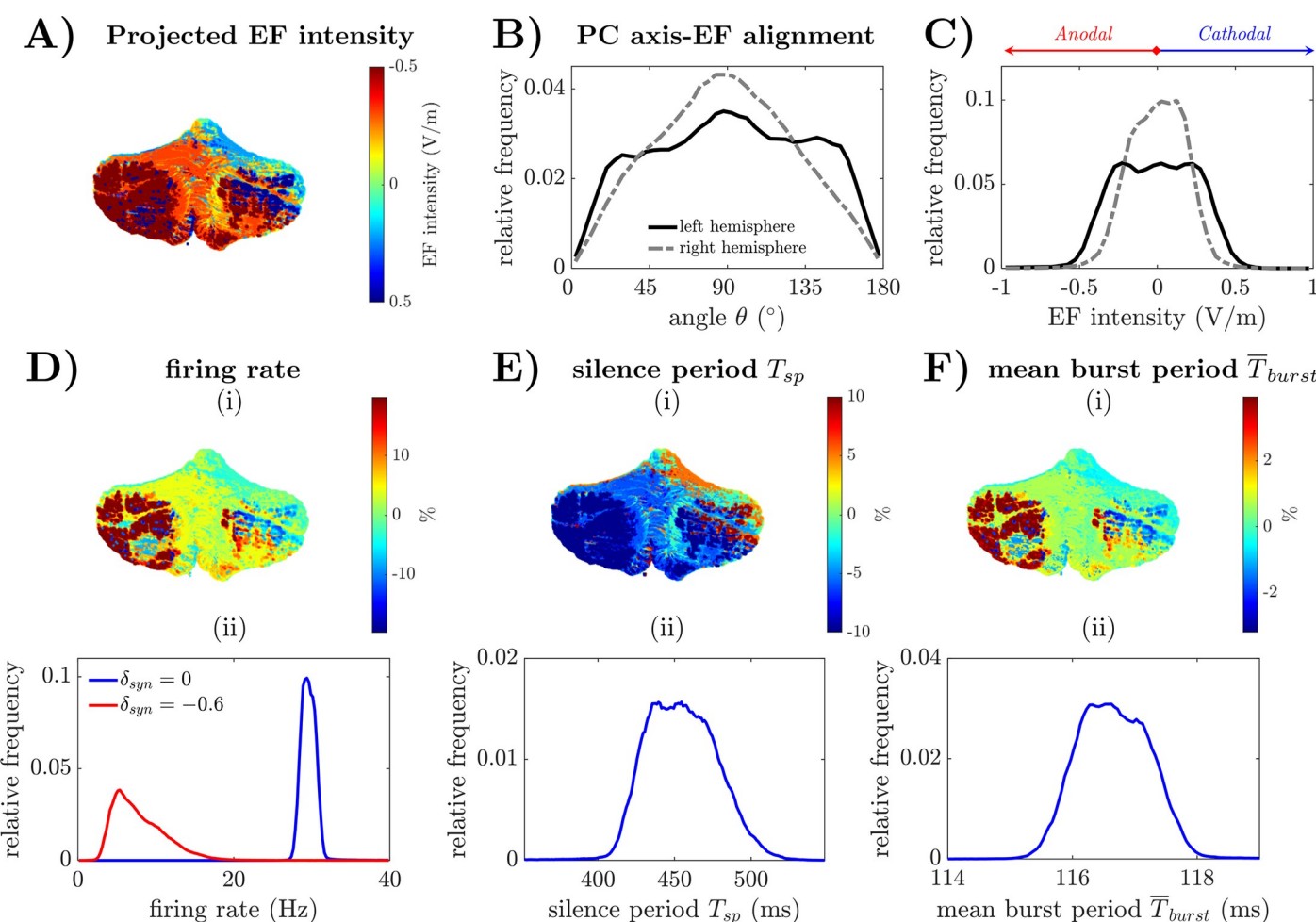

**Fig 8. Distribution of EF-induced variations of PC spiking across the cerebellar cortices.** (A) Distribution of the EF projection onto the somato-dendritic axis of Purkinje cells along the cerebellar cortical surface for one subject, i.e., atlas 5 in [26]. Camera view azimuth and elevation angles are -90° and 0°, respectively. (B) Sample distribution of the angle $\theta$ between the PC somato-dendritic axis and the EF along the cerebellar cortical surface (*gray line*: right hemisphere; *black line*: left hemisphere). The tDCS pad electrode was placed on the left cerebellar hemisphere. (C) Distribution of the EF projections on the somato-dendritic axis of Purkinje cells for the left hemisphere (*black line*) and the right hemisphere (*gray line*), respectively, under anodal R-tDCS. Anodal EF intensities (*red arrow*) and cathodal EF intensities (*blue arrow*) are indicated. (D-F) Distribution of mean firing rate under presynaptic activation ($\delta_{syn} = 0$) (D), silence period, $T_{sp}$ (E), and mean burst duration, $\overline{T}_{burst}$ (F), respectively, of the Purkinje cells along the cerebellar cortical surface. Panels (i) in (D-F) show the spatial mapping of the estimated values on the cerebellar cortical surface of atlas 5 (azimuth: -90°; elevation: 0°). Panels (ii) in (D-F) report the sample distribution of the estimated metrics across four subjects. In (D), panel (ii) reports the distribution of the mean firing rate for $\delta_{syn} = 0$ (*blue line*) and $\delta_{syn} = -0.6$ (*red line*), respectively. Silence period in (E) and mean burst periods in (F) are calculated as in Figs 6 and 7, respectively.

depolarized than hyperpolarized. The opposite scenario would occur, instead, by placing the cathodal electrode on the cerebellum.

Based on the values of the EF locally projected onto the Purkinje cells, we estimated the spatial distribution of the average firing rate (FR) of the PC soma under polysynaptic activation. We also estimated the distribution of the post-CS silence period, $T_{sp}$, and average burst period, $\overline{T}_{burst}$. **Fig 8D, 8E and 8F** report the variation of FR, $T_{sp}$, and $\overline{T}_{burst}$ from the baseline values (i.e., values estimated under no stimulation) throughout the cerebellar cortex when 2-mA R-tDCS is applied. Across four subjects, we found that, under nominal synaptic gain value (i.e., $\delta_{syn} = 0$) the stimulation caused a significant departure of the average FR from the baseline value, **Fig 8D (i)** (average variation: 5.68 ± 0.78%, mean ± S.D.; one-sample *t*-test, *P*-value $P<0.0001$), with approximately 9.91%, 1.47%, and 1.07% of Purkinje cells reporting a change (either

**Table 3. Percentage of Purkinje Cells Under Field Intensity Exceeding 0.5 V/m and 1.5 V/m.** Percentage of Purkinje cells under projected EF intensities exceeding 0.5 V/m and 1.5 V/m in each atlas and averaged across all atlases, mean ± S.D.

| | | Projected EF Intensity | | | |
|---|---|---|---|---|---|
| | | <-0.5V/m | >+0.5V/m | <-1.5V/m | >+1.5V/m |
| **Atlas no.** | **1** | 3.67% | 1.17% | 1.64% | 0.11% |
| | **2** | 3.35% | 1.31% | 1.28% | 0.06% |
| | **4** | 0.48% | 0.56% | 0.48% | 0.00% |
| | **5** | 2.85% | 0.68% | 1.06% | 0.01% |
| | **Average** | 3.04±0.60%** | 0.93±0.36% | 1.12±0.48%** | 0.045±0.053% |

** indicates that the difference from the opposite field direction (i.e., >+0.5 V/m vs. <-0.5 V/m; >+1.5 V/m vs. <-1.5 V/m) is statistically significant (two-sample t-test, P-value, P<0.005). Positive and negative field intensities indicate cathodic and anodic stimulation, respectively.

increase or decrease) of more than 10%, 20% and 30%, respectively, across the entire cerebellum, **Fig 8D (ii)** (*blue line*). In case of stronger synaptic inhibition (i.e., $\delta_{syn}$ = -0.6), instead, tDCS resulted in a wider range of FR values, i.e., 3–24 Hz, and caused a significant shift in the average FR from the baseline (average variation: 95.1 ± 23.8%, mean ± S.D.; one-sample *t*-test, P-value P<0.0001), with 69.2% of Purkinje cells reporting a change (either increase or decrease) of more than 30%. A similar trend was reported for $T_{sp}$, **Fig 8E (i)**, and $\overline{T}_{burst}$, **Fig 8F (i)**, even though the maximum variation was smaller, i.e., ±10% and ±4%, respectively. Moreover, the percentages of Purkinje cells that exhibited a reduction or an increment of $T_{sp}$ higher than 10% and $\overline{T}_{burst}$ higher than 2% were similar ($T_{sp}$: 6.01% ± 1.40%; $\overline{T}_{burst}$: 0.79% ± 0.24%), **Fig 8E** and **8F (ii)**, thus resulting in a limited average variation across the cerebellum (average change: 1.28% ± 0.38% and 0.16 ± 0.00% for $T_{sp}$ and $\overline{T}_{burst}$, respectively, mean ± S.D.).

Altogether, these results indicate a net anodal polarization across the cerebellar cortex, with a skewed distribution of higher EF intensities toward the region directly underneath the electrode. This distribution (i) primarily affects the discharge pattern of the Purkinje cells that receive strong synaptic inhibition, (ii) causes a significant shift in the spiking mode across the entire cerebellar cortex, and (iii) results in a variation of the average firing rate that may be, in localized regions of the cortex, as high as 90% of the pre-stimulation baseline value, depending on the underlying level of synaptic activity. Moreover, the changes in silence period and the small but steady variation of the average burst period suggest that 2-mA-tDCS can evoke a significant, polarization-dependent modulation of the activity of Purkinje cells *in vivo*, which can directly influence the mechanisms of neural encoding and synaptic plasticity controlled by the spiking of the Purkinje cells.

## 3. Discussion

The relationship between the morphology of neurons and their sensitivity to applied EF has been extensively investigated in the cerebral cortex [12,13]. Moreover, tools used to investigate the polarization of cortical neurons have been recently extended to estimate the membrane polarization of Purkinje cells in cerebellum [42]. Nonetheless, a quantitative analysis of the effects of transcranial stimulation on cerebellar neurons is lacking.

In healthy individuals, 2-mA-tDCS of the cerebral cortex is expected to induce a membrane polarization of 1 mV or less in pyramidal cells and interneurons [11]. Such polarization values are known to induce long-term potentiation plasticity and determine a strong modulation of the corticospinal excitability [67], which indicate that the acute effects of tDCS on the cerebrum likely involve the network connectivity of the cortex [14]. The cerebellum, though, has a different cytoarchitecture and fewer recurrent connections compared to the cerebrum [68,69],

and this suggests that the sensitivity of the cerebellar neurons to tDCS can hardly be inferred from the characterization of the cerebral cortex under constant electric fields.

Furthermore, while pyramidal neurons are mildly responsive to low-intensity transcranial alternating EF [70], it has been shown in anesthetized rats that alternating EF can entrain Purkinje cells into bursting, with the burst frequency coinciding with the frequency of the alternating field [44]. Finally, *in vitro* studies have shown that the major cerebellar neurons (i.e., Purkinje cells and deep cerebellar neurons) can exhibit a wide range of discharge patterns in response to synaptic stimuli, and the transition from one type of discharge pattern (e.g., bursting, regular simple spiking, etc.) to one another can occur rapidly in response to electric fields of moderate amplitude [43]. Overall, this suggests that, despite the modest intensity of the induced electric fields, cerebellar tDCS can induce a substantial modulation of the neuronal activity in cerebellum *in vivo* which needs to be fully characterized.

In this study, we propose a computational pipeline to assess the effects of cerebellar tDCS on key parameters and spiking behaviors that uniquely characterize the neuronal activity in cerebellum. We show that, even though tDCS-induced EF can reach deep cerebellar nuclei, the effects of polarization are mainly confined to Purkinje cells along the cerebellar cortex. Deep cerebellar neurons, in fact, show modest changes to the firing rate and rebound spiking under tDCS, even though the field-induced membrane polarization is like in Purkinje cells, while no significant polarization is reported for granule cells. This is consistent with previous evidence supporting the role of the Purkinje cells as the primary target of tDCS [8] and indicates that the small size of the cells in the granular layer [71] and the robustness of the deep cerebellar neurons against frequency adaptation in the presynaptic stimuli [72] are likely responsible for the weak response to tDCS-induced polarization in deep structures of the cerebellum.

Moreover, our simulations indicate that cerebellar tDCS affects the excitability of the Purkinje cells at rest and can modulate how Purkinje cells respond to synaptic inputs *in vivo*. Depending on the polarity, tDCS-induced EF modulate the firing rate at rest by primarily altering the somatic transmembrane voltage and shifting the timing of the action potentials as the spiking voltage threshold is approached. This suggests that tDCS can directly affect the intrinsic excitability of the Purkinje cells [73] and therefore amplify the neuronal responsiveness [74]. Also, when synaptic stimuli are applied, the tDCS-evoked shift in firing pattern grows as the gain of the synapses increases. Inhibition to the Purkinje cells mainly comes from the molecular layer interneurons and is considered the primary mechanism through which the cerebellar output is controlled with high spatiotemporal precision [51,75]. Hence, the fact that the firing pattern of Purkinje cells may become sparse or tonic because of low-intensity EF (i.e., ±1.5V/m) suggests that tDCS may significantly alter the precision through which the cerebellar output is controlled by the interneurons, thus impacting the cerebellar response to conditional stimuli and therefore the motor learning process, e.g., see [76] for a recent review of the issue.

Studies *in vitro* and *in vivo*, e.g., [53,77,78], have shown that the spiking pattern of Purkinje cells also depends on the morphology of the dendrites, which would suggest that the effects of the electric fields may be cell-specific. However, the morphology of the dendritic arborization varies modestly across Purkinje cells [78] and mainly affects the average firing rate [79]. Hence, even though morphology-related changes to the Purkinje cell's firing activity are expected, it is plausible that the application of electric fields will result in discharge patterns that fall within the classification and class-transition map reported in our study.

Finally, our findings suggest that cerebellar tDCS may significantly alter bursting and complex spiking in Purkinje cells, and these alterations are predicted to be (i) related to the amplitude and orientation of the applied electric field and (ii) primarily mediated by somatic ionic channels, i.e., $Na^+$ channels in case of silence period and $K^+$ channels in case of bursting,

respectively. Although the ionic channel composition of Purkinje cells has recently gone through revision and still remains under debate, e.g., [18], previous studies have clarified that simple and complex spikes initiate in the proximal axon of the Purkinje cells and then back-propagate to the soma [80]. Hence, our simulations indicate that, even though the soma does not initiate spikes in Purkinje cells, the polarization of the soma can ultimately be responsible for controlling the interval between consecutive bursts of action potentials.

This is particularly relevant in the case of complex spikes. Complex spiking is considered relevant to conditioning [60,81], and the silence period following a complex spike is crucial for the rebound activation of the deep cerebellar neurons, which form the primary output structure of the cerebellum [82]. Hence, the modulation of the silence period may directly impact the conditioned responses. This is consistent with earlier studies investigating the impact of tDCS on motor learning [83–86] and suggests that the facilitation of the conditioned eyeblink response, which has been reported under anodal tDCS [83,84], may be associated with a reduction in the silence period following complex spikes.

It should be emphasized that, while the magnitude of the response of Purkinje cells to transcranial stimulation may be remarkable, the portion of Purkinje cells that can effectively report a large shift in activity under tDCS is overall small and localized in the cortical area around the location of the cerebellar electrode. In fact, we show that the convoluted architecture of lobules and gyri forming the cerebellar cortex may result in an ample range of orientations of the Purkinje cells with respect to the direction of the applied EF, and this can result in an inconsistent response to tDCS across contiguous Purkinje cells, even within small regions. Interestingly, we show that the proportion of Purkinje cells that aligned in the direction of the EF is similar to the proportion of cells that are against the direction of the induced field, and this may explain why anodal and cathodal tDCS over the cerebellum are often reported to exert identical effects, especially for large electrode pads, e.g., [29,85,87].

It should be pointed out, however, that even though the cell populations aligned with and against the electric field could be similar in size, Purkinje cells located in the area directly underneath the electrode polarized mainly according to the tDCS-evoked electric field and received far greater EF intensity than the other cerebellar regions. Also, several traits of the neuronal response to tDCS, e.g., the silence period and the instantaneous firing rate, showed a significant lack of symmetry as the field varied from anodal to cathodal values, with larger magnitude under anodal stimulation than cathodal stimulation. This skewness may account for the tDCS polarity-specific effects observed in several studies about motor learning and posture control, e.g., [8,84,86,88].

Finally, we acknowledge that, while our study has mainly focused on the direct effects of tDCS on post-synaptic neuronal activity, there is *in vitro* evidence that tDCS can also modulate presynaptic activity and synaptic transmission, e.g., [32,89]. Hence, it is plausible that the magnitude of the effects of tDCS on cerebellar neurons may exceed our model predictions. This is particularly true in case of deep cerebellar neurons, which receive direct projections from the Purkinje cells, whose high sensitivity to tDCS has been predicted in our study. It is less likely that tDCS would modulate the activity of climbing fibers and mossy fibers, which are responsible for the presynaptic activity towards Purkinje cells and granule cells, respectively, mostly because of the deeper location and lack of dendritic arborization of these fibers. As well, the parallel fibers, which are located in the molecular layer and provide diffused glutamatergic synapses onto the Purkinje cells, are mostly arranged perpendicularly to the direction of the electric fields. Hence, although possible, it is unlikely that tDCS would induce noticeable changes to the presynaptic excitation provided by the parallel fibers to the Purkinje cells.

### 3.1 Translational implications for clinical applications

Cerebellar tDCS has been recently proposed as a neuromodulation therapy to treat motor dysfunctions in ataxia [2,3], focal dystonia [90], and essential tremor [91,92], but the mechanisms of action remain unclear, and the titration of the stimulation has resulted remarkably challenging thus far. Our study provides a characterization of the acute effects of tDCS on cerebellar neurons and shows that the effects can both recapitulate and reconcile a diverse range of clinical results recently reported in the literature. More importantly, our study indicates that, depending on the placement of the electrode on cerebellum, the consistent polarization of the Purkinje cells underneath the electrode region may provide a bulk effect that can directly percolate to the cerebellar nuclei. Since abnormal oscillations and pathological cerebellar activity are expected to mediate the manifestation of motor disfunctions [53,93,94], our study contributes to understand the mechanisms through which tDCS may alter these abnormal activity patterns, thus enabling the treatment of movement disorders.

Clinically, severe neurological conditions of various etiology have been linked to a maladaptive change of the excitability and firing pattern of Purkinje cells. For instance, a dysregulation of the Purkinje cells' postsynaptic activity has been shown to lead to the motor symptoms of restless leg syndrome [95], vestibulo-ocular reflex impairment [96], and paroxysmal kinesigenic dyskinesia-induced dystonia [97]. Moreover, a diffused reduction of the Purkinje cells' excitability throughout the cerebellar cortex has been shown to impair cerebellar zonal patterning and contribute to cerebellar atrophy [98,99], which are both associated with impaired motor learning, loss of coordination, and ataxia. Vice versa, recent optogenetic work showed that a selective inhibition of Purkinje cells of the vermis could help control temporal lobe seizures [100].

Altogether, these studies suggest that the excitability of the Purkinje cells can be a key control factor in the formation of severe symptoms across a wide range of chronic neurological conditions. Accordingly, our findings indicate that cerebellar tDCS can be a useful therapeutic option for several conditions beyond those for which tDCS is currently tested. Moreover, our study points out that, by carefully designing the field intensities and electrode shape and configuration, tDCS can selectively modulate the excitability of the Purkinje cells in specific zones of the cerebellum. Although speculative at the moment, this could have therapeutic effects in seizure control and modulation of nonmotor circuits.

### Methods

Numerical calculations of the EF induced by cerebellar tDCS montages in the brain were conducted in ROAST ver. 3.0 [27]. Numerical simulations of multicompartment computer models of the cerebellar neurons were conducted in NEURON, ver. 7.7.2 [33] with CNEXP ode solver and 0.025 ms integration step. Simulation results were analyzed in MATLAB, rel. 2017a, The MathWorks, Inc., Natick, MA. Neuron models and simulation scripts are available in ModelDB, URL: http://modeldb.yale.edu/267189.

### Modeling the EF generated by cerebellar tDCS

The 3T MRI atlases from human subjects considered in this study were reported in [26] (atlas 1, 2, 4, and 5) along with segmentation masks to isolate the cerebellar regions. Atlas 3 was not included due to insufficient tissue contrast around the cerebellar areas. The voxel resolution of the brain scans was resampled from 300 μm (original value in [26]) to 600 μm to reduce the computational cost. The R-tDCS montage was simulated with one electrode (anode) centered at location E133, EGI HCGSN-256 EEG system, Electrical Geodesics, Inc., Eugene, OR, and one supraorbital electrode (cathode) centered at location E18. Electrode locations were chosen

as in [2] to apply unilateral stimulation over the left cerebellar hemisphere. See [29] for comparison with electrode configurations with cathode centered on the buccinator or deltoid muscles. The current intensity of each electrode was set as 2 mA.

The HD-tDCS montage, which is used to facilitate motor adaptation, was simulated as in [101]. Specifically, we adopted a 10–20 system with five ring electrodes of the same size, with the anodal electrode on the inion (Iz) and four cathodal electrodes in position Oz, O2, P8, and PO8, respectively. The current intensity was set at 2 mA for the anode and 0.5 mA for each cathode electrode.

For each combination of brain scan, tDCS montage, and current intensity, we first segmented the brain images into six tissue categories (**S1 Table**) and built the volumetric mesh of the reconstructed brain via tetrahedral meshing [102] (MATLAB function **iso2mesh()**). Then, we solved the Laplacian equation $\nabla^2 v = 0$, where $v$ is the electric potential, at the nodes of the mesh by using the open-source solver getDP [103] with boundary conditions given by the current intensity at the anode and cathode. Finally, we assigned the EF to every voxel as $\overrightarrow{E} = \nabla v$, and we extracted the voxels that span the cerebellar regions by applying the segmented masks provided in [26]. All lobules were combined as the cerebellar cortex, and the white matter was labeled as the cerebellar nuclear region [104].

## Cerebellar neuron models

We considered 3D multicompartment models of PC, GrC, and DCN because these cell types account for more than 80% of cerebellar neurons [105]. Although relevant in shaping the firing rate of Purkinje cells [51], cerebellar interneurons were omitted because they are significantly less numerous than granule cells [106], smaller than Purkinje cells [21], and with a limited dendritic span. Furthermore, interneurons have low firing rate at rest (e.g., ~5 Hz *in vivo*, [107]) and have shown modest responses to low-intensity EF [43]. Golgi cells were also omitted because they target only granule cells and have therefore no direct contact with Purkinje cells [108]. Finally, Golgi cells and interneurons are scarcely arranged within the molecular layer of the cerebellum [109].

The PC model was adopted from [34], consists of 1,611 compartments, and provides a 3D reconstruction of the morphology of a whole Purkinje cell obtained from a 2–3-month-old Guinea pig. Fifteen types of ionic channels were included in the PC model, including Na$^+$ channels Nav1.6, potassium channels Kv1.1, 1.5, 3.3, 3.4, and 4.3, and calcium channels KCa1.1, 2.2, and 3.1. The corresponding ionic channel mechanisms were modeled from published references without modifications of their kinetics. See [34] for details on compartment dimensions, ionic channel models, distribution of channels across compartments, and tuning of free parameters.

The DCN model was adopted from [35], consists of 516 compartments, and provides a 3D reconstruction of the morphology of a deep cerebellar neuron obtained from a 13-to-19-day-old Sprague-Dawley rat. Nine types of ionic channels were included in the DCN model, including fast and persistent Na$^+$ channels, K$^+$ channels Kv2 and Kv3, purely Ca$^{2+}$-gated K$^+$ channels, and Ca$^{2+}$ channels Cav3.1. Ionic channel kinetics were modeled after published analyses of DCN neuron conductance, and free parameters were estimated by fitting *in vitro* DCN voltage responses to current injection pulses via genetic algorithms. See [35] for details on compartment dimensions, ionic channel models, distribution of channels across compartments, and tuning of free parameters.

The GrC model consists of 65 compartments and was created for this study by combining the morphology of a whole human granule cell and the ionic channel modeling proposed in [36]. The morphology (NMO_32569) includes soma and dendritic arborization and was

reconstructed from a neurologically normal, 54-year-old male who had died of acute myocardial infarction [110]. We processed the cell morphology using the CellBuilder tool in NEURON [33] to obtain the somatic and dendritic compartments, see **S2 Table**. We then extended the morphology by adding the hillock (5 compartments), and an axon perpendicular to the somato-dendritic plane, including the axonal initial segment (AIS, 30 identical compartments) and ascending axon (AA, 4 compartments). The dimensions of the hillock, AIS, and AA were chosen as in [36]. Nine types of ionic channels were assigned to dendrites, soma, hillock, and axon as in [36], and include fast, persistent, and resurgent $Na^+$ channels, N-type $Ca^{2+}$ channels, voltage-activated $K^+$ channels, and $Ca^{2+}$-gated $K^+$ channels. Kinetics of the ionic channel models were adopted from [111,112], while the passive electric properties and the ionic channel maximum conductance values were optimized in NEURON (PRAXIS optimization method) to fit the *I-f* (i.e., injected current vs. spike frequency) curve reported in [36], see **S3 Table**. For all three models, the temperature was set as 37˚C wherever applicable. Models were simulated when no EF was applied and for EF of intensity ranging from -1.5 V/m to +1.5 V/m (13 values).

## DCN rebound firing

We added 450 GABAergic synapses to the DCN model (one synapse per dendritic compartment and 50 synapses on the soma [35]), and all synapses were simulated in NEURON as Net-Con objects [33]. Briefly, every synapse responded to a presynaptic pulse with a postsynaptic current

$$I_{syn}(t) = g_{syn}\alpha(t - \tau)(V - E_{syn}), \tag{1}$$

where $\tau$ is the arrival time of the presynaptic pulse, $V$ is the membrane potential at the postsynaptic dendritic compartment to which the synapse is attached, and $\alpha(z) = A(e^{-z/\tau_2} - e^{-z/\tau_1})$ is the activation function, with $A$ chosen such that $0 \leq \alpha(z) \leq 1$ for all $z$. The maximum conductance $g_{syn} = 1.89 \times 10^{-3}$ µS and the time constants $\tau_1 = 0.18$ ms and $\tau_2 = 3.61$ ms are as in [113], and the reversal potential $E_{syn}$ = -90 mV was as in [35] to further hyperpolarize the neuron, thus inducing strong rebound bursts afterwards.

To mimic the activation of a presynaptic Purkinje cell, all synapses on the DCN model were driven by a common train of pulses, and the arrival times of the pulses were irregularly spaced by drawing the inter-pulse intervals, $T_{syn}$, from an exponential function, i.e., $T_{syn} \sim t_r + \text{Exp}(1/\lambda)$, with $\lambda = 2$ ms and $t_r = 2$ ms, where $t_r$ accounts for the refractory period between spikes. The presynaptic train started at 300 ms, spanned 15 ms (i.e., the typical duration of a PC complex spike [24]), and was designed to simulate the arrangement of the CS spikelets [114].

The after-hyperpolarization (AHP) instantaneous firing rate (IFR) histogram was constructed as in [35] to quantify the rebound firing activity. The AHP IFR was obtained as convolution of the somatic spikes with a Gaussian kernel, whose S.D. is $\sigma_k = \min(ISI_{before}, ISI_{after})/\sqrt{2\pi}$. The values $ISI_{before}$ and $ISI_{after}$ are the inter-spike interval (ISI) preceding and immediately following the spike, respectively. AHP IFR histograms were estimated when no EF is applied and under ±1.5 V/m EF. The difference between the AHP IFR histograms estimated under EF and at baseline (i.e., no EF applied) was measured throughout the AHP rebound phase, i.e., until the firing rate at the DCN soma returned to the pre-hyperpolarization level.

## GrC relay fidelity to mossy fibers

We added one AMPA synapse and one NMDA synapse on the most distal compartment of the longest dendritic branch. Synapses were modeled as in (1) with $g_{syn} = 2.88 \times 10^{-3}$ µS

(AMPA) and $g_{syn} = 4.51\times10^{-4}$ μS (NMDA) and remaining parameters $\tau_1 = 0.6$ ms, $\tau_2 = 1.0$ ms, and $E_{syn} = 0$ mV. The number, type, location, and conductance of the synapses were chosen as in [47] to mimic the adaptation of the GrC response to mossy fiber inputs.

Both synapses were driven by a regular train of pulses, and the frequency of the train, $f_{stim}$, was varied from 5 Hz to 140 Hz (9 values). For each value of $f_{stim}$, the GrC model was simulated for 2,300 ms, with the train starting 100 ms after the simulation onset. The last 2,000 ms of GrC activity were considered to estimate the fidelity relay ratio, $r$, i.e., $r$ was the ratio between the number of spikes at the GrC soma and the number of pulses delivered in the 2,000-ms-window.

## PC response to synaptic stimuli

We placed synapses across the dendritic tree of the PC model, and we measured the response of the PC soma to the selective activation of these synapses. The following three configurations of synaptic stimuli were implemented.

**Scenario 1.** A single current pulse (duration: 0.5 ms, amplitude: 0.5 nA) was applied at the most distal dendritic compartment (i.e., 305.5 μm from the soma), and the response of the PC was measured at the soma. This scenario aims to measure the transmission delay, $T_d$, i.e., the lag between the input at the dendrite and the resultant action potential at the soma. Simulations lasted 2,000 ms, and the current pulse was applied at time $t = 500$ ms. To correctly isolate the somatic response, the PC model was prevented from spiking spontaneously at rest by setting the membrane potential to -65 mV and blocking $Na^+$ and $Ca^{2+}$ channels in the axonal initial segment.

**Scenario 2.** We placed a glutamatergic synapse on the PC dendritic compartment considered in *Scenario 1* and let the PC model exhibit tonic spiking at rate $f$ between 50 Hz and 70 Hz at rest. The synapse was modeled as in (1) ($g_{syn} = 6.014\times10^{-3}$ μS, $\tau_1 = 0.6$ ms, $\tau_2 = 1.0$ ms, and $E_{syn} = 0$ mV, [52, 77]) and was activated with a random train of presynaptic pulses that mimic weak synaptic stimuli from parallel fibers. Presynaptic pulses arrived with inter-pulse intervals $T_{syn}$ randomly generated according to $T_{syn} \sim \text{Exp}(1/\lambda)$, with $\lambda = 50$ ms. The PC model was simulated for 100,000 ms, and presynaptic pulses started at time $t = 2,000$ ms.

We considered the PC as an oscillator with oscillation period $1/f$, and we used the sequence of action potentials elicited at the soma in response to the presynaptic train to estimate the PC phase response curve (PRC) as in [50]. The PRC describes how the phase of an oscillation is affected by the phase at which a perturbation is delivered and predicts conditions for oscillators like Purkinje cells to synchronize or desynchronize.

**Scenario 3.** In this scenario, glutamatergic (from parallel fibers) and GABAergic (from cerebellar interneurons) synapses were concurrently activated over time and resulted in an irregular spiking pattern at the PC soma. The activation of both excitatory and inhibitory synapses mimicked the condition of Purkinje cells *in vivo* during motor execution [55,56]. We evaluated both the change in the firing rate and the evolution of spiking patterns as the synaptic intensity was varied.

We placed 1,000 glutamatergic synapses and 100 GABAergic synapses on randomly selected, non-overlapping compartments of the PC dendritic tree, and every synapse responded to a train of presynaptic pulses. The ratio between glutamatergic and GABAergic synapses was adjusted to reproduce the excitation/inhibition ratio of Purkinje cells [115]. The postsynaptic current resulting from the combination of all synapses was given by the formula:

$$I_{syn}(t) = \sum_{j} g_{glut} \sum_{r_j} \alpha(t - \tau_{r_j})(V - E_{glut}) - \sum_{k} g_{GABA} \sum_{r_k} \alpha(t - \tau_{r_k})(V - E_{GABA}) \tag{2}$$

where the function $\alpha(\cdot)$ is as in (1), $\tau_{r_j}$ is the arrival time of the presynaptic pulse $r_j$ to the

glutamatergic synapse $j$ ($1 \leq j \leq 1,000$) and $\tau_{r_k}$ is the arrival time of the presynaptic pulse $r_k$ to the GABAergic synapse $k$ ($1 \leq k \leq 100$). For the GABAergic synapses, $\tau_1$ and $\tau_2$ in $\alpha(\cdot)$ and $E_{GABA}$ were set to 1.0 ms, 5.0 ms, and -80 mV, respectively, [77]. For the glutamatergic synapses, $\tau_1$, $\tau_2$, and $E_{glut}$ were set to the values used in *Scenario 2*. The maximum conductance $g_{glut}$ and $g_{GABA}$ were constant and assigned to each compartment.

For the baseline condition, $g_{glut} = \hat{g}_{glut} = 2.5 \times 10^{-4}$ μS and $g_{GABA} = \hat{g}_{GABA} = 1 \times 10^{-3}$ μS, where these values were chosen from [116] and [115], respectively. Both values were then varied between 40% and 160% of the baseline value with increments of 20%, i.e., we considered the values $g_{syn} = (1 + \delta_{syn})\hat{g}_{syn}$ ($syn = glut$ or $GABA$) for $-0.6 \leq \delta_{syn} \leq 0.6$. The arrival times $\tau_{r_j}$ and $\tau_{r_k}$, instead, varied across the synapses as every synapse was driven by a different train of exponentially spaced pulses as in *Scenario 2*, with $\lambda = 100$ ms for glutamatergic synapses and $\lambda = 35.7$ ms for GABAergic synapses. In this way, the PC model fires irregularly at ~30Hz at baseline with a coefficient of variance (CoV) of the inter-spike interval CoV = 0.8 [77]. Also, the PC firing rate significantly depends on $\delta_{syn}$ and ranges from a few spikes per second to ~50Hz as $\delta_{syn}$ increases from -0.6 to +0.6, thus replicating a range of physiological behaviors reported for PC.

Three sets of synaptic placements and presynaptic pulse trains were randomly generated, resulting in three configurations. For each configuration, the PC model was simulated for 10,000 ms, and the PC firing rate at the soma was averaged across configurations. EF (when applied) started at $t = 2,000$ ms and lasted till the end of the simulation.

## PC complex spiking and spontaneous bursting

To simulate the activation of climbing fibers, we adopted the configuration proposed in [18], i.e., we placed glutamatergic synapses on all dendritic compartments with a diameter $d \geq 3.5$ μm, including the dendritic trunk (105 compartments altogether), we modeled the postsynaptic current at each synapse as in (1), with $g_{syn} = 1.59 \times 10^{-3}$ μS, $E_{syn} = 0$ mV, $\tau_1 = 0.3$ ms, and $\tau_2 = 3.0$ ms, and we drove all synapses with a common presynaptic pulse applied at time $t = 500$ ms, thus resulting in a complex spike at the soma followed by silence period. The silence period $T_{sp}$ was defined as the interval between the largest spike in the complex spike and the first simple spike, i.e., a spike occurring as part of the regular self-sustained somatic spiking pattern, after the prolonged hyperpolarization.

Purkinje cells also exhibit spike adaptation and transient bursting in response to the activation of the granular layer both *in vitro* and *in vivo* [23]. To assess this bursting activity, we simulated the *in vitro* experiment conducted in [25], i.e., we injected a 2-nA-depolarizing step current into the soma of the PC model at time $t = 200$ ms to trigger a transient burst pattern. Bursting was then measured in terms of burst count, burst period $T_b$, and mean burst rate. For any burst $k$, $k = 1, 2, 3, \ldots,$ $T_b$ was defined as the interval between the last spike of burst $k$ and the last spike of the following burst, i.e., burst $k+1$. The "*burst interval*" associated with burst $k$, instead, was defined as the interval between the last spike of burst $k$ and the first spike of the following burst, and the "*spiking period*" of the burst was the interval between the first spike and the last spike of the burst. In our analyses, a burst is a sequence of action potentials with less than 10 ms between any two consecutive action potentials.

## Spatial distribution of tDCS-induced modulation of Purkinje cell firing

The effects of tDCS on PC firing (i.e., firing rate, silence period, and average burst period) were mapped onto the cerebellar cortex according to the following steps. The procedure was repeated for each subject (i.e., atlas) individually, and results were averaged across atlases.

*Step 1.* We isolated the cerebellar cortex by setting a threshold on the voxel intensity (threshold is 1,300 for atlas 1–2 and 1,100 for atlas 4–5). We processed voxels spanning the surface of the cerebellar cortex and up to 1.2 mm in depth, which is the region where Purkinje cells are located. For each voxel, we calculated the EF intensity and orientation in ROAST.

*Step 2.* For each voxel, we determined the orientation of the somato-axonal axis and somato-dendritic axis of the Purkinje cells in the voxel. The somato-axonal axis was assumed perpendicular to the voxel outer surface. For the somato-dendritic axis, instead, we observed that Purkinje cells have dendritic trees growing perpendicularly to the folds of the cerebellar cortex [117,118]. Hence, we determined the iso-surfaces of the image using the MATLAB function **isosurface()** (*isovalue* parameter set as 0.5) and we computed the correspondent normal vectors with the MATLAB function **isonormals()**, which provided the orientation of the PC dendritic trees. Normal vectors on the white matter were removed because they correspond to the cerebellar nuclei, which do not contain Purkinje cells [104].

*Step 3.* For each voxel and iso-surface of interest, we used the EF intensity in the voxel and the relative orientation of the EF with respect to somato-dendritic and somato-axonal axes along the iso-surface to estimate the EF projection along the PC somato-axonal axis. We then interpolated the sensitivity curves in **Figs 5–7** at the intensity of the EF projection and determined the specific value of firing rate, silence period, and burst period expected for the Purkinje cells along the iso-surface in the assigned voxel.

## Supporting information

**S1 Fig. Distribution of the EF intensity across the cerebellar volume in atlas 1, 2, 4.** (A-C) Sample distribution of electric field (EF) intensity estimated for atlas 1 (A), 2 (B), and 4 (C) from [26], respectively, under regular (R)-tDCS with two pad electrodes. R-tDCS montage is as depicted in **Fig 1A**. Panels (i) and (ii) in (A-C) report the EF intensity distribution for the left (target) cerebellar hemisphere (i) and the right cerebellar hemisphere (ii), respectively. For each hemisphere and atlas, probability functions are estimated separately for voxels mapping the cerebellar cortex (*blue lines*) and the cerebellar nuclei (*red lines*). These EF intensity distribution functions complement the presentation of the results in **Figs 1A** and **2A**. Sample probability distribution functions were computed as in **Fig 2A**.
(TIF)

**S2 Fig. EF intensity across the cerebellar volume in case of tDCS with two ring electrodes.**
(A) Estimated intensity and orientation of the EF induced by 2-mA-cerebellar regular (R)-tDCS delivered with two ring electrodes in the brain for one human subject, i.e., atlas 5 from [26]. Panel (i) reports the position of the anode and cathode. Panels (ii), (iii), and (iv) report an axial, coronal, and sagittal view of the EF distribution. Colormap in (i) indicates the distribution of the electric potential, *v* (scale on the left). In (ii)-(iv), colormaps indicate the EF intensity, and black arrows indicate the EF orientation. Color scale in (iii) also applies to (ii) and (iv). (B) Sample distribution of the EF intensity for the R-tDCS montage in (A). Sample distributions are reported for the left (target) cerebellar hemisphere (i) and the right cerebellar hemisphere (ii), respectively. Distribution functions are computed separately for voxels mapping the cerebellar cortex (*blue lines*) and the cerebellar nuclei (*red lines*). EF intensity in the left hemisphere is higher compared to the right hemisphere, both for the cortex and the nuclei (one-way ANOVA test after Bonferroni correction, *P*-value $P<0.001$). Results in this figure can be directly compared with the results in **Figs 1** and 2. Graphs in this figure were generated as reported for **Figs 1** and 2.
(TIF)

**S3 Fig. Presynaptic stimuli to Purkinje cells.** (A) Normalized scalogram of the weighted sum of 1,000 glutamatergic and 100 GABAergic synaptic activation sequences applied on the dendrites of the PC model to generate the results in **Fig 5**. The summed sequence is: $S = g_{GABA}(60 + E_{GABA})h_{GABA} - g_{glut}(60 + E_{glut})h_{glut}$, where $g_{glut}$, $E_{glut}$, and $h_{glut}$ are the synaptic conductance, reversal potential, and time histogram (bin size: 1 ms) of the activation sequence for the glutamatergic synapses, respectively; $g_{GABA}$, $E_{GABA}$, and $h_{GABA}$ are the synaptic conductance, reversal potential, and time histogram for the GABAergic synapses, respectively. The scalogram was computed via continuous wavelet transform (Morse kernel, sampling frequency: 1,000 Hz), filtering with a 30-Hz low-pass filter, and averaging samples over time. (B) Coefficient of variance (CoV) of the inter-spike intervals estimated at the PC soma for different combinations of values for parameter $\delta_{syn}$ and R-tDCS-induced EF intensity, $E$. Pairs ($\delta_{syn}$, $E$) resulting in *dense* and *irregular* spiking as defined in the main text are characterized by low CoV (i.e., <2; below the opaque green plane) while pairs resulting in *bursting* and *sparse* spiking are characterized by high CoV values (>2; above the opaque green plane). CoV was not computed for $\delta_{syn} = -0.6$ and $E = 0.75$ or $\delta_{syn} = -0.6$ and $E = 1.5$V/m due to too few spikes in the entire spike train. (C) Transmembrane voltage at the PC soma (*black lines*) and estimated total synaptic current (*red lines*) under tDCS-induced EF at -1.5 (i), -0.75 (ii), 0 (iii), 0.75 (iv), and 1.5 V/m (v), respectively, and $\delta_{syn} = -0.4$. Negative and positive EF intensities correspond to anodal and cathodal stimulation, respectively. Total synaptic current (right vertical axis) is estimated for -60-mV-membrane potential. Negative values indicate that the total synaptic current is hyperpolarizing. The results presented here complement the analysis reported in **Fig 5**.
(TIF)

**S4 Fig. Projection of the EF onto the cerebellar cortices for different electrode configurations.** (A) Spatial distribution of the projections of the electric field (EF) onto the somato-dendritic axis of the Purkinje cells along the cerebellar cortical surface in three subjects, i.e., atlas 1 (i), atlas 2 (ii), and atlas 4 (iii) from [26], respectively. EF is generated for regular 2-mA-tDCS with pad electrodes, i.e., same configuration as in **Fig 1A**. (B) Spatial distribution of the EF projections onto the somato-dendritic axis of the Purkinje cells along the cerebellar cortical surface of one subject, i.e., atlas 5 from [26], in case of regular tDCS with two ring electrodes (i) and high-density tDCS with five electrodes (ii), respectively. Montages in (i) and (ii) are as in **Figs S2A** and **1B**, respectively. Color scale on the right applies to all plots in (A-B). These plots complement the results reported in **Fig 8**.
(TIF)

**S5 Fig. Effects of segmentation on the orientation of cerebellar surfaces.** Sample distribution function of the angle $\varphi$ between a preset vector pointing 45° between the anterior and superior axes towards the cerebellar midline and the normal vector on the cerebellar surface for four subjects (i.e., atlas 1, 2, 4, 5 from [26]) and the sample high-resolution atlas (*Ref*) presented in [119].
(TIF)

**S1 Table. Conductivities assigned to tissues and layers in ROAST.** The table reports the conductivity assigned in ROAST to different neural tissues, skin, bone, electrodes used for transcranial stimulation, and conductive gel on the electrodes. Conductivities are assigned as in [120].
(XLSX)

**S2 Table. Electrotonic compartments in the granule cell (GrC) model.** The table reports the sections of the GrC model along with their number, diameter, length, capacitance ($C_m$), and cytoplasmic axial resistivity ($R_a$). For sections including more than one compartment, values

are reported as mean ± S.D. unless the same value was used for all compartments. The total length of the dendritic tree, axon initial segment, and ascending axon is 158.6 μm, 70.1 μm, and 218.0 μm, respectively.
(XLSX)

**S3 Table. Ionic mechanisms in the GrC model.** The table reports the location, maximum conductance ($g_{max}$), and reverse potential ($E_{rev}$) for all ionic channels. Ionic channel models, i.e., mechanisms and gating equations, are as in [36]. AIS = axonal initial segment. AA = ascending axon.
(XLSX)

**S4 Table. Variation in PC model's somatic ionic charge density vs. field intensity.** The table reports the absolute difference ($\Delta c$) between the charge density estimated for ionic currents measured at the soma of the PC model under constant electric fields (EF) of intensity -1.5 V/m and +1.5 V/m, respectively. Charge density of an ionic current is measured at the soma during the silent period. Ionic currents include leakage currents (leakage) and currents flowing through sodium ($Na^+$) Nav1.6 channels, hyperpolarization activated cyclic nucleotide-gated cationic (HCN) channels, potassium ($K^+$) channels (combination of Kv1.1, Kv1.5, Kv3.3, Kv3.4, Kv4.3, KCa1.1, KCa 2.2, KCa 3.1, and Kir2 channels), and calcium ($Ca^{2+}$) channels (i.e., combination of Cav3.1, Cav 3.2, Cav 3.3, and Cav2.1 channels). See [34] for a definition of the ionic channel mechanisms. Since the duration, $T_{sp}$, of the silent period changes with the EF intensity, the charge density was computed as the integral of the ionic current density from the end of the complex spike till the end of the shortest $T_{sp}$, i.e., 251 ms.
(XLSX)

**S1 Text. Supplementary Methods.** The text describes the procedures used to generate the graphs in S5 Fig and assess the effects of segmentation on the orientation of the cerebellar surfaces.
(DOCX)

## Author Contributions

**Conceptualization:** Xu Zhang, Roeland Hancock, Sabato Santaniello.

**Data curation:** Xu Zhang.

**Formal analysis:** Xu Zhang.

**Funding acquisition:** Sabato Santaniello.

**Investigation:** Xu Zhang.

**Methodology:** Xu Zhang, Roeland Hancock, Sabato Santaniello.

**Project administration:** Sabato Santaniello.

**Resources:** Roeland Hancock, Sabato Santaniello.

**Software:** Xu Zhang, Roeland Hancock.

**Supervision:** Roeland Hancock, Sabato Santaniello.

**Validation:** Xu Zhang.

**Visualization:** Xu Zhang, Sabato Santaniello.

**Writing – original draft:** Xu Zhang.

**Writing – review & editing:** Roeland Hancock, Sabato Santaniello.

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
