## [Decision Letter · Decision Letter 0]

21 May 2021

Dear Dr. Santaniello,

Thank you very much for submitting your manuscript "Transcranial Direct Current Stimulation of Cerebellum Alters Spiking Precision in Cerebellar Cortex: A Modeling Study of Cellular Responses" for consideration at PLOS Computational Biology.

First of all, I want to apologize that it has taken so long to come back to you with an evaluation of your manuscript. As with all papers reviewed by the journal, your manuscript was reviewed by members of the editorial board and by several independent reviewers. It has been incredibly difficult to find expert reviewers for your work, which is why it took so long. At this point, however, we are fortunate to have received reviews of three experts and in light of the reviews (below this email), we would like to invite the resubmission of a significantly-revised version that takes into account the reviewers' comments. 

We cannot make any decision about publication until we have seen the revised manuscript and your response to the reviewers' comments. Your revised manuscript is also likely to be sent to reviewers for further evaluation.

Sincerely,

Lore Thaler

Guest Editor

PLOS Computational Biology

Kim Blackwell

Deputy Editor

PLOS Computational Biology

Reviewer's Responses to Questions

**Comments to the Authors:**

Reviewer #1: In the paper of Santaniello et al., the authors investigate the effects of transcranial direct current stimulation in cerebellum using MRI and modelling of electric fields and multi-compartment models of neurons. It proposes cerebellar tDCS as a therapeutic option for disorders involving cerebellar hyperactivity such as ataxia.

Overall, the paper is well written. The main objectives and findings are clearly laid out. The findings are interesting and are well explained to the reader. Below are my specific comments:

Major points:

Three multicompartment neurons models have been used in this study and one of them has a simple morphology (GrC). Why does GrC have a simple morphology compared to the other ones? I would suggest having additional realistic morphologies (or to have three full complex morphologies at least). Moreover, we are not given many details about these morphologies (especially for the intrinsic dynamics).

Some figures (Fig1 and 2) are quite difficult to read clearly (too many subpanels) and/or we are given too much information not necessarily pertinent. I would suggest keeping key information/graph. For example, no need of the 3 polarization patterns on Figure 1.C(i-iii). Just one would be enough to illustrate the neuron polarization.

Minor points:

Line 85: add the reference radman2009cortical.

Line 150: finding higher electric field intensity in the cerebellar nuclei than in in the cerebellar cortex was expected?

Line174: How much lower? It is significant but giving some numbers would be appreciated.

Line 185: previous studies have introduced the polarization length. Explain it in short sentence and add references.

Line 199: the firing rate of some cells was not highly affected by the tDCS was also demonstrated by previous studies.

Line 205: give some reasons to explain the difference of polarization of cells (morphologies…)

Line 246: the firing rate value. Is there a specific reason to have a firing rate of ~100Hz?

Line 264: linear relation. What is the correlation value? Coefficient of determination R2?

Line 295: How is the threshold of 80ms chosen? Same question for the 80-200ms range. Add reference for these values.

Line 307: units of the slope.

Line 314: ‘overall limited… within +/- 25%’. This number is quite a lot for a firing rate variation so talking about ‘limitation’ is not entirely correct.

Line 327: Has this soma transition been observed in previous studies (modeling or in vitro)?

Line 368: Is there a general method to characterize complex spikes? (threshold, number of spikelets, time latency…)

Line 396: Why specifically focusing on the Na+ v1.6 dynamic? Did previous study investigate a relation between the silent period and this channel? Or more generally, the ionic channels? If yes, add reference.

Line 453: results with the dendritic polarization. The authors claim that ‘the dendritic polarization […] modestly changed the burst count and the mean period” ; on the figure 4.E and 4.F, the green line shows almost no difference with the amplitude of E – especially for the burst count. We could see a light decrease of the T_burst for amplitude above 0.6 though.

Line 518: distribution between 30 and 150degres. Give a percentage of the distribution (70%, 80% are in this range?)

Line 520: concerning the supp. Fig S5, why can we observe 2 symmetric spikes around 45 and 135degres?

Line 578: pyramidal cells are known for being quite responsive to electric field (tACS or tDCS) and more responsive than inhibitory cells or interneurons. Even for low amplitude (~1mV/mm)

Line 609: the firing pattern not only depends on the intensity of electric field but also on the morphology of the cell or the ionic channels. It would be interested to investigate these parameters in Purkinje cells. Any idea on the influence of these parameters?

Line 652: add a paragraph about the clinical application of the findings and how it can be related to brain disorders.

Line 700: 3D multi-compartment models have ben used but we don’t know many details about them. Give more details about the dynamics (active soma (Hodgkin Huxley?), passive dendrite.…), ionic channels modeling.

Line 704: Cerebellar interneurons and Golgi cells have been omitted for some reasons, but it would be interesting to see how they are affected by the electric field (in supplementary with preliminary results or in the next paper).

Line 712: Why was the range +/- 1.5 mV/mm chosen? Is there any specific reason? Because the stronger electric field is, the stronger effects we will observe.

Line 731: a GABAergic synapse is added to every dendritic compartment of the DCN model. How many are there?

Line 739: E_syn=-90. How was this value chosen? Add reference.

Line 808: add reference for these values.

Minor points on figures:

All figures: the font of the caption for each figure should be the same font as in the main body. Increase the font size too and the space.

Fig 1.B: So, there are 3 different types of cells (PC, GrC, DCN). What does the ‘~700um’ or ‘~20um’ refer to? The GrC does not have a realistic morphology (just a ball-and-stick and 4 dendrites) compared to the PC and DCN. Why?

Fig 1.B.i: What is ‘NOR1, NOR2 and NOR3’?

Fig 1.C: add an arrow to show the direction of the electric field.

Fig 2.C: what is the distance of the distal dendrite of the soma?

Fig 2.D: For φ ~ 0.6, we have almost a perfect superposition of the 3 curves. How can you explain that? And why after that point, the red curve is under the black one and the blue line above the black one?

Fig 2.E.ii: what is the method to distinguish dense and irregular spiking?

Fig 2.E.i: What is the ratio between excitatory / inhibitory synapses? Add it in the main text. Any reference?

Reviewer #2: Summary :

- Combined MRI-derived reconstruction of the cerebellum with a tDCS-induced EF.

- Multicompartmental cerebellar neurons : Purkinje cell, deep cerebellar neuron, granule cell.

- tDCS significantly modulates the post-synaptic spiking precision of PCs: firing pattern, timing

- No significant influence on DCN and GRC.

- tDCS modulated the synaptic inhibition (observed with discharge pattern, average firing rate, spiking pattern)

- The acute effects of tDCS on the cerebellum mainly focus on Purkinje cells and modulate the precision of the PC response to synaptic stimuli, thus having the largest impact when the cerebellar cortex is active.

- Method : FEM, HD-tDCS & pad-tDCS, 2mA, ROAST

Strong Point: considering the fact that most previous studies reported cortical neurons’ response to tDCS, this research may be novel.

Weak Point: Overall, there is a lack of explanation or interpretation of results (give a reason and explanation with a citation). The implication of the result only is described. In addition, methodology was quite poorly explained (for example, two pad stimulation was applied, however, detailed head model used here and configuration of electrodes were not presented. Authors addressed that ROAST was used only.

This study calculated a realistic EF range and apply the realistic EF to a multi-compartmental neuron model with a quasi-uniform assumption. This means there was a lack of information on the spatial distribution of neurons in a cerebellum, although the angle between EF and dendrite-soma-axon axis was considered.

Comparing to multi-scale modeling, this approach is not realistic or advantageous.

Detailed Comments

1. The authors mentioned higher EF intensity found under the source electrode. However, there is a lack of explanation of why EF intensity in cerebellar nuclei is higher than that in the cerebellar cortex. Higher EF is observed with a greater distance between electrodes (Parazzini et al 2012, Shahid et al 2014)

2. Pad and ring electrode shows distinctive EF distribution, thus, it is better to include a supplementary figure1 in another figure ( not a supplementary)

3. Insufficient explanation of why GrC and DCN were not affected by tDCS as much as PC.

4. Result part

- Cerebellar tDCS induces mild electric fields across the cerebellum part.

- P.8 177-9. Please cite previous studies and mention consistency with a previous study’s result of electric field distribution of HD-tDCS.

- Fig1 b ii) why only GrC’s morphology is different from other neurons? It is better to give unity to all neuronal models’ figures.

- For validation of the neuron model, a linear increase in membrane polarization with the field intensity is important. However, it is hard to capture a linear increase in polarization through Table1 and explanation in the result and discussion part. It is better to use a plot or show increasing values in the table.

5. Method part

- Please provide a specific method used to calculate EF in ROAST.

- Please address information about head models (for example in Fig. 1A)

- A definition and method of polarization length and sensitivity are are omitted. Please describe it with a citation.

- Please provide a specific method of how to pair (couple) EF with multi-compartmental neuronal models.

- Please provide a brief summary of how to calculate extracellular potential, instead of just citing [79].

- If the authors used the ‘extracellular mechanism’ of NEURON, calculating potential is required. But, the way of calculating potential was omitted. Please address it.

6. Additional comments

- In figure 4A, burst period, burst interval and spiking period need to be defined with mathematical term to eliminate ambiguity.

- In figure 5A, the intention of the arrangement of the figures is not clear.

Reviewer #3: In this timely study the authors use MRI-derived reconstruction of cerebellum to 1) calculate field intensities across the cerebellum and 2) quantify the immediate impact of cerebellar tDCS on three different types of cerebellar neurons by using multi-compartments models of this neurons. The manuscript is very well written and the figures are clear and very informative. The authors focused in the impact of tDCS on different neuronal parameters, as PC firing, complex spike wavelets or post-CS silence with important physiological consequences. I appreciate that authors clearly state the model’s prediction on each one of the different sections.

I have some general and minor comments and suggestions that could improve the manuscript.

General Comments

Although authors clearly state that, they only focus on PC, DCN and GrC neurons, this last group of cell (GrC) are almost missed along result sections. Considering its relevant role in the cerebellar network activity and its very characteristic morphology (ascending axon and parallel fibers) it is expected that mild changes could have strong functional impact in cerebellar processing. Although silent at rest, could exogenous electric field modulate its firing rate during sensory activation for example?

tDCS has been shown to modulate presynaptic activity in-vitro and in alert animals… How could these presynaptic effects modify your model predictions?

Minor comments

- The parcellation of the cerebellum that authors use in this study seems to have more detail than the head model. How the authors got the information about the direction of the E-field with respect to the cortical surface? When mapping the E-field direction, can the authors guarantee they're not making any errors and assigning E-field values from the CSF (for instance) to the cerebellum parcel?

- Regarding the methods authors use to create the head model: Did authors correct manually the segmentations? Did they check the results of the segmentation?

**Have all data underlying the figures and results presented in the manuscript been provided?**

Reviewer #1: Yes

PLOS authors have the option to publish the peer review history of their article (what does this mean?). If published, this will include your full peer review and any attached files.

Reviewer #1: No

Reviewer #2: No

Reviewer #3: **Yes: **Javier Márquez-Ruiz

**Have the authors made all data and (if applicable) computational code underlying the findings in their manuscript fully available?**

Reviewer #2: Yes

Reviewer #3: Yes
---

## [Decision Letter · Decision Letter 1]

2 Nov 2021

Dear Dr. Santaniello,

We are pleased to inform you that your manuscript 'Transcranial Direct Current Stimulation of Cerebellum Alters Spiking Precision in Cerebellar Cortex: A Modeling Study of Cellular Responses' has been provisionally accepted for publication in PLOS Computational Biology.

Best regards,

Kim T. Blackwell, V.M.D., Ph.D.

Deputy Editor

PLOS Computational Biology

Kim Blackwell

Deputy Editor

PLOS Computational Biology

Reviewer's Responses to Questions

**Comments to the Authors:**

Reviewer #3: The authors have satisfactorily answered all my questions and incorporated the main suggested changes into the text.

In my opinion the manuscript is ready to be accepted.

**Have the authors made all data and (if applicable) computational code underlying the findings in their manuscript fully available?**

Reviewer #3: Yes

PLOS authors have the option to publish the peer review history of their article (what does this mean?). If published, this will include your full peer review and any attached files.

Reviewer #3: **Yes: **Javier Márquez-Ruiz

---

## [Editor Report · Acceptance letter]

6 Dec 2021

PCOMPBIOL-D-20-02326R1 

Transcranial Direct Current Stimulation of Cerebellum Alters Spiking Precision in Cerebellar Cortex: A Modeling Study of Cellular Responses

Dear Dr Santaniello,

I am pleased to inform you that your manuscript has been formally accepted for publication in PLOS Computational Biology. Your manuscript is now with our production department and you will be notified of the publication date in due course.

With kind regards,

Anita Estes
